# Proteasomes accumulate in the plant apoplast where they participate in microbe-associated molecular pattern (MAMP)-triggered pathogen defense

Hana Zand Karimi [1,3], Kuo-En Chen [1], Marilee Karinshak[1], Xilin Gu [2], Jason K. Sello [2] & Richard D. Vierstra [1] ✉

Akin to mammalian extracellular fluids, the plant apoplastic fluid (APF) contains a unique collection of proteins, RNAs, and vesicles that drive many physiological processes ranging from cell wall assembly to defense against environmental challenges. Using an improved method to enrich for the *Arabidopsis* APF, we better define its composition and discover that the APF harbors active proteasomes though microscopic detection, proteasome-specific activity and immunological assays, and mass spectrometry showing selective enrichment of the core protease. Functional analysis of extracellular (ex)-proteasomes reveals that they help promote basal pathogen defense through proteolytic release of microbe-associated molecular patterns (MAMPs) such as flg22 from bacterial flagellin that induce protective reactive-oxygen-species (ROS) bursts. Flagellin-triggered ROS is also strongly suppressed by the enigmatic *Pseudomonas syringae* virulence effector syringolin-A that blocks ex-proteasome activity. Collectively, we provide a deep catalog of apoplast proteins and evidence that ex-proteasomes participate in the evolving arms race between pathogens and their plant hosts.

The extracellular space is an important microenvironment within cellular organisms that includes all domains outside the plasma membrane toward the external surroundings. In plants, this sizable compartment, commonly known as the apoplast, encompasses the porous cell wall, extracellular spaces, and xylem, all of which are bathed in apoplastic fluid (APF)[1]. While first thought to be largely inert, the APF is now known to serve numerous functions essential to cell wall synthesis and remodeling, cell-to-cell and longer-range communications, nutrient transport, environmental sensing and response, microbiome refugia, and as a first-line defense against invading viral, bacterial, and fungal pathogens[2,3]. It presumably does so through a diverse collection of metabolites, proteins, RNA, lipids, extracellular vesicles (EVs), hormones and other signaling molecules, and microbial byproducts, many of which are poorly cataloged at present[4,5]. That the composition of the APF can vary dynamically belies active transport processes which can be regulated by development, environmental perturbation, and pathogenic and/or symbiotic microbe/host interactions[2]. For example, recent studies have shown that plants actively secrete tiny/short, circular, and long-noncoding RNAs complexed with RNA-binding proteins either in free or EV-encapsulated forms into the APF for microbial defense[6–8]. Connections between APF pH and composition and either development or salt/drought stress have also been seen in multiple plant species[9–12]. As the APF remains one of the least understood plant compartments, defining its makeup and functions is of emerging importance.

[1]Department of Biology, Washington University in St. Louis, St. Louis, MO, USA. [2]Department of Pharmaceutical Chemistry, University of California, San Francisco, CA, USA. [3]Present address: Pfizer Pharmaceuticals, Chesterfield, MO, USA. ✉e-mail: rdvierstra@wustl.edu

Among the numerous enzymes found in the plant APF are proteases, including members of metallo, aspartyl, cysteinyl, and subtilisin-type serine protease families[4,5,13,14], which could be vital to various physiological processes such as cell wall remodeling, protein turnover, zymogen activation, intercellular signaling, and biotic protection. Notably, several proteases along with other hydrolytic enzymes accumulate in the apoplast near sites of pathogen invasion[15,16], suggesting that these catabolic activities are crucial safeguards within the plant innate immune system. In particular, apoplastic proteases have been implicated in pathogen-triggered immune (PTI) responses elicited by microbe-associated molecular patterns (MAMPs) proteolytically derived from conserved microbial proteins[15,17,18]. The resulting immunogenic MAMP peptides are recognized by dedicated host pattern recognition receptors, which then trigger a coordinated defense response.

Arguably, the best understood MAMP response involves the flg22/FLS2 signaling system[17,18]. Here, the flagellin glycoprotein, which constitutes the primary structural element of bacterial flagella, including those from various plant pathogens such as *Pseudomonas syringae*, is digested by host glucanases and proteases to release the naked 22-amino-acid flg22 peptide. While the full panoply of proteases that generate MAMPs such as flg22 is unknown, the apoplastic subtilases SBT5.2 and SBT1.7 have been recently implicated[19]. Released flg22 is then detected by the plasma membrane-bound, pattern recognition receptor FLS2, which initiates a complex defense response upon engagement that includes a burst of reactive oxygen species (ROS) at the infection site[18,20]. This ROS then auto-propagates as a wave among adjacent cells to trigger a general stress-protective response[21]. It is noteworthy that some host proteases are targeted by pathogen-secreted inhibitors to suppress immune signaling[22–24], thus underscoring their importance to a co-evolving host/pathogen arms race involving proteolysis. An intriguing example is the non-ribosomal peptide syringolin-A (SylA) which is secreted by virulent *P. syringae* strains to enhance infectivity possibly by inhibiting host proteasomes selectively[25,26].

Surprisingly, besides various proteases[13], proteomic studies with several plant species have hinted at the presence of proteasomes in the APF[4,9,12,14,27,28], which provide the intracellular proteolytic activities central to both ubiquitin (Ub)-dependent and Ub-independent protein breakdown. Its 20S core particle (CP) assembles as a C2-symmetric barrel that houses the protease active sites, which is often capped at one or both ends with an asymmetric 19S regulatory particle (RP) to generate an ATP-dependent 26S complex directed toward degrading ubiquitylated substrates[29,30]. While these extracellular (ex)-proteasomes were dismissed as cytosolic contaminants of prior plant APF preparations, recent studies with various mammalian extracellular fluids have detected both the CP with its associated peptidase activities, and sometimes the 26S holocomplex capped with the RP[31]. Investigations into these ex-proteasomes found them to be uniquely modified[32], and suggested roles in degrading external proteins, including those associated with various pathologies[33,34].

During the course of our transmission electron microscopic (TEM) characterizations of the *Arabidopsis* APF[6], we noticed particles morphologically similar to the CP with its four stacked heptameric rings creating a distinctive barrel with axial pores. Here, we further investigated this possibility by coupling an improved APF isolation protocol that included ATP to help stabilize possible 26S complexes[35,36], together with a collection of proteasome-focused assays, including TEM, tandem mass spectrometry (MS/MS), immune-depletion, and peptidase activity and inhibitor studies, to demonstrate that the *Arabidopsis* APF does indeed contain functional ex-proteasomes. While this fluid is mostly populated with the CP, 26S particles singly or doubly capped with the RP were also evident.

Anticipating that these ex-proteasomes participate in MAMP-triggered immune defense, we tested for such a role using the flg22/FLS2 system and found that they help induce ROS bursts upon *P. syringae* infection likely by facilitating flg22 release from bacterial flagellin. Both the activity of ex-proteasomes and flagellin-induced ROS could be effectively suppressed by SylA, thus defining how this proteolytic toxin enhances *P. syringae* infection. Collectively, this study provides an improved catalog of APF-resident proteins, confirms the accumulation of proteasomes outside of plant cells, and discovers a crucial role for ex-proteasomes in pathogen defense by helping create immunogenic epitopes central to PTI signaling.

## Results

### Enrichment of APF from *Arabidopsis* leaves

Given the persistent challenges in identifying bona fide apoplast-resident proteins apart from cytosolic contaminants, we adapted the vacuum-infiltration approach[37] to better isolate the *Arabidopsis* APF (Fig. 1a). It involved harvesting mature leaves by cutting just up from the stem/petiole junction, and infiltrating them with a mild salt

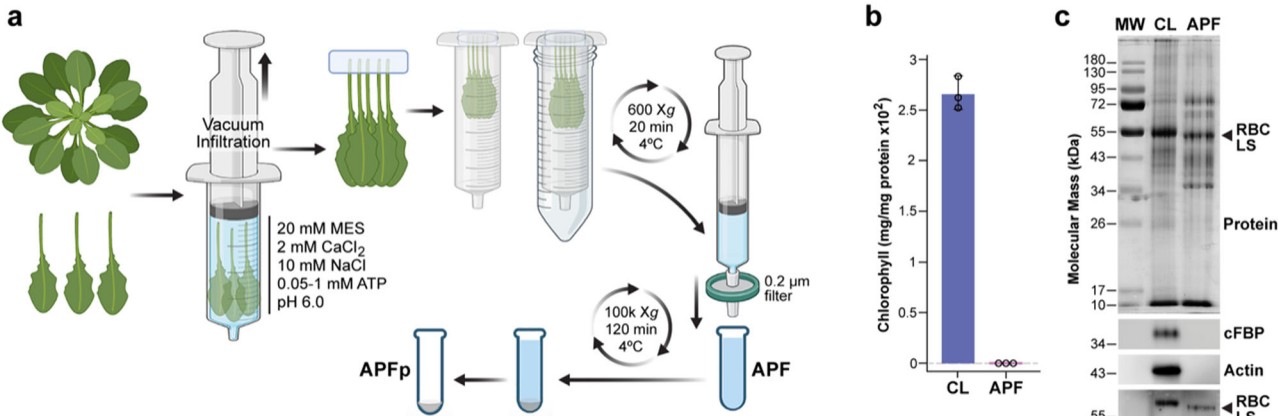

**Fig. 1 | Isolation protocol and characterization of the APF from *Arabidopsis* leaves. a** Protocol for isolating the leaf APF. Non-damaged 5-6-week-old leaves were vacuum infiltrated with extraction buffer containing ATP under a mild negative pressure with the infiltrate then collected by low *g* centrifugation of vertically hung leaves. Clarification of the fluid using a 0.2-μm cut-off filter generated the APF, whose vesicles and large particles were then concentrated if needed by centrifugation at 100,000 X*g*, resulting in the APF pellet (APFp). Created in BioRender. Karimi, H. (2025) https://BioRender.com/r81f161. **b** Level of chloroplast contamination in the APF as compared to the crude leaf lysate (CL) by spectrophotometric assays for chlorophyll. Bars reflect the mean (±SD) of three technical replicates. Individual data points are included. **c** The distribution of protein, cytosolic FBP, actin, and the Rubisco large subunit in the CL and APF by staining for protein with silver or by immunoblot analysis with specific antibodies. MW, molecular mass markers.

extraction buffer under low negative pressure to minimize cell/tissue damage. Given that the plant APF typically contains micromolar concentrations of ATP[38,39] and that ATP stabilizes 26S proteasome assembly and subsequent activity[35,36], we included 0.05 to 1 mM ATP in this buffer. The leaves were hung with their petioles up to avoid fluids leaking from the cut site, and then centrifuged under a low $g$ force to collect the infiltrate presumably exiting through stomata and hydathodes. This fluid was clarified with a 0.2-μm cutoff filter to remove large debris thus generating the APF. In some studies focusing on possible ex-proteasomes, we then concentrated large particles from this dilute APF by centrifugation at 100,000 X$g$ to generate the APF pellet (APFp)[6–8], which was resuspended in extraction buffer (Fig. 1a).

Using chlorophyll as a marker for chloroplast contamination, we found little, if any, in the APF (<1%) as compared to crude lysates (CL) generated by direct homogenization of leaves followed by a low $g$ clarification (Fig. 1b). Similarly, little contamination (<1%) was seen by immunoblotting the APF for cytosolic fructose bisphosphatase (cFBP) and actin (Fig. 1c) as well as by the distinct SDS-PAGE patterns of total protein in the APF versus CL (Fig. 1c). Nonetheless, we routinely detected varying amounts of chloroplast-localized Rubisco in the APF by immunoblotting with antibodies against the chloroplast-encoded ~55-kDa large subunit (Fig. 1c); it migrated as a slightly lower apparent molecular mass species during SDS-PAGE whose origin is not yet understood (see below).

## Proteomic Analysis of the APF

As a fourth purity test, we compared the proteomes of the APF and CL by liquid chromatography (LC)-MS/MS after trypsinization. The presence of each protein was verified based on four biological replicates each analyzed by two technical replicates; inclusion in our stringent lists required detection of at least one peptide in each of the four biological replicates. In total, 2925 proteins were cataloged in the CL and 1380 proteins in the APF when aligned against the *Arabidopsis* proteome database (http://www.Arabidopsis.org) (Supplementary Data 1 and 2). While 1156 proteins were shared amongst the CL and APF datasets, another 219 proteins were uniquely detected in the APF (Fig. 2c; Supplementary Data 3).

Subsequent comparisons of the lists were challenged by two complications, with one being correct compartment assignments. While most intracellular predictions were reasonably accurate for *Arabidopsis* as assigned by The *Arabidopsis* Information Resource (TAIR; https://www.Arabidopsis.org), those for the apoplast were not. As a solution, we generated a catalog of likely APF candidates by aggregating previous proteomic datasets from *Arabidopsis* samples enriched in either the apoplast, extracellular spaces, or the cell wall[5,11,40], along with apoplast designations found in TAIR. The composite list was then culled for proteins known or previously predicted to reside in other compartments (e.g., Rubisco and its activase, ribosomal proteins, proteasome subunits, histones, PEP carboxylase, and proteins integral to photosynthetic light capture; 29 in total (see Supplementary Data 4) to generate a final catalog of 3127 predicted apoplast polypeptides, of which ~20% were previously designated as "unclassified" in the Gene Ontology (GO) database (http://geneontology.org), and thus might represent new apoplastic constituents (Supplementary Data 5).

A second challenge to dataset comparisons was the dominance of chloroplast proteins in both numbers and apparent abundance in the *Arabidopsis* leaf CL samples, which filtered into APF samples as likely contaminants. For example, while only 32.2% of the CL proteome detected here was predicted to be of chloroplast origin by our GO classifications, we estimated that these proteins actually accounted for 53.3% of protein abundances using the combined ion intensities determined from the MS1 scans as a semi-quantitative measure (Fig. 2a; Supplementary Fig. 1a). In fact, 19 of the top 30 CL proteins based on MS1 ion intensities were from chloroplasts, with another 14

also being chloroplastic when analyzing the next 27 proteins (58% in total); the remainder were assigned to other compartments by GO (e.g., peroxisomes, mitochondria, cytoplasm, nuclei, ER, vacuoles, and apoplast) (Supplementary Data 1). Moreover, the two most abundant proteins in *Arabidopsis* leaves – large and small subunits of Rubisco – had combined MS1 ion intensity values (2.91 e + 12 and 1.38 e + 12, respectively) roughly 8 times higher than the next protein on the list – SALICYLIC ACID BINDING PROTEIN3/β-carbonic anhydrase (2.62 e + 11), which agreed with their robust accumulation in photosynthetic tissues.

These chloroplast protein levels dropped substantially to 15.5% for the APF samples based on combined MS1 ion intensities (Fig. 2a) in line with the purity markers we used to assess non-APF compartments (Fig. 1b, c). While Rubisco large and small subunits were still abundant, only 26 of the first 78 proteins (33%) in the APF list were assigned to chloroplasts by GO, with 44 expected to be apoplastic (56%) based on our updated apoplast catalog (Supplementary Data 2). To avoid bias caused by an overabundance of Rubisco, we then removed its large and small subunits from subsequent proteomic analyses of the APF. Here, enrichment values for the APF changed significantly from being 29% and 26.7% of the proteins in the dataset being assigned to the apoplast and chloroplasts, respectively, based on protein numbers (Supplementary Fig. 1a) to being 70.3% and 15.5%, respectively, based on abundance estimates derived from the combined MS1 ion intensities after adjustment (Fig. 2a). Further analyses of the MS1 ion-intensity data by Pearson's correlation and $R^2$ values comparing Rubisco levels to those of the top 20 CL and APF proteins were also consistent with Rubisco behaving as a variable contaminant in the APF preparations.

Volcano plots for proteins in common between the CL and APF lists (1156 total) further illustrated this point. As can be seen in Fig. 2b, those proteins significantly enriched in the CL were mostly chloroplastic (277 of 362 total; 76.5%) based on $Log_2$ fold change (FC) $\geq 1$ or $\leq -1$ and $P$-values in significance <0.05, while predicted apoplastic proteins were now significantly enriched in the APF (102 of 269 total; 37.9%). GO analyses illustrated by pie charts for other cytosolic compartment/complexes further emphasized our minimization of intracellular contaminants in the APF. With the exception of the plasma membrane, both actual numbers and combined MS1-ion intensity values for ER, mitochondrial, peroxisomal, and nuclear proteins all dropped in the APF versus the CL (35% and 66% less, respectively), with few nuclear proteins detected in the APF (Fig. 2a; Supplementary Fig. 1a).

This strong depletion of chloroplast, mitochondrial, and peroxisomal but not plasma membrane proteins in the APF was also evident in targeted volcano plots comparing the CL and APF fractions (Supplementary Fig. 1b). This slight enrichment of plasma membrane proteins in the APF versus the CL might reflect increased contamination due to their proximity to the apoplast, and/or recent findings that a subset of EVs arise from fusion of intracellular multivesicular bodies with the plasma membrane before secretion[5].

Comparative GO analyses of the CL, APF, and APF-specific proteins further confirmed the strong enrichment for apoplastic proteins using our improved APF isolation method. As expected, CL assignments were strongly biased toward chloroplasts based on Cellular Compartment, Biological Process, and Molecular Functions GO classifications, with chloroplast, and thylakoid being top terms for Cellular Compartment, and various enzymatic categories related to photosynthesis enriched in the other two categories (Fig. 2c). By contrast, GO analyses of the APF identified the apoplast, extracellular, and secretory vesicles as top terms for Cellular Compartments, with various catalytic activities associated with the cell wall, carbohydrate metabolism, hydrolase, and defense terms being foci for the Biological Process and Molecular Functions categories (Fig. 2c). Notably, the preference for apoplast functions was even stronger when comparing

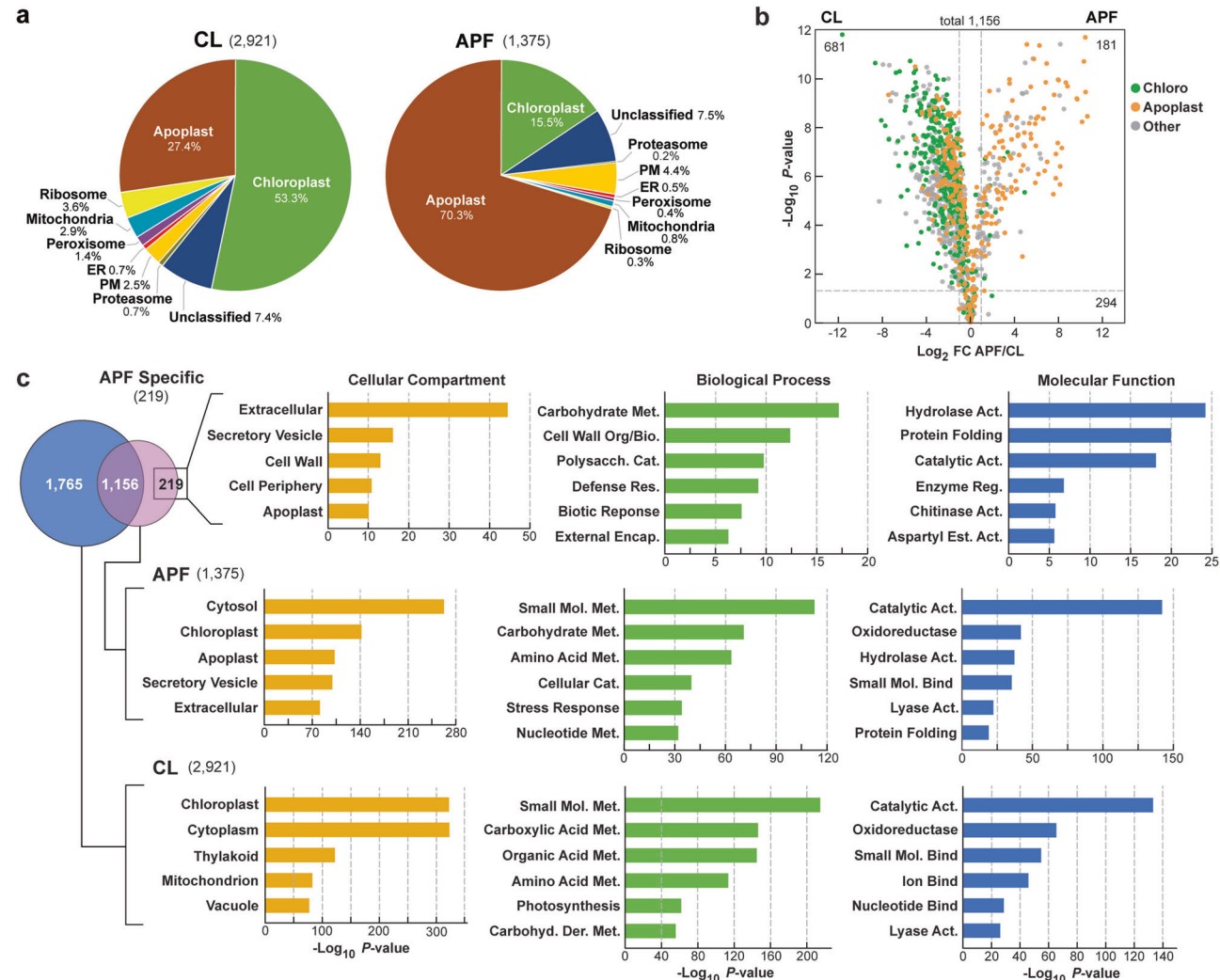

**Fig. 2 | Proteomic analysis of the *Arabidopsis* leaf APF.** Total protein from the clarified CL and the APF were trypsinized and subjected to LS-MS/MS. Individual proteins were identified by Proteome Discover whose abundances were then determined semi-quantitatively based on the combined ion intensities determined from the MS1-scans (e-values). The CL and APF were each assayed by four biological replicates each measured with two technical replicates; protein abundances were then normalized based on the total MS1 ion intensities for all proteins in each. Only those proteins detected in at least one technical replicate for each of the four biological replicates were included. Values for the Rubisco large and small subunits were removed from the APF prior to analysis to avoid their abundance bias. **a** Pie charts showing the percentage of proteins localized to specific compartments/complexes in the APF versus CL using assignments either provided by GO or from a

master list of apoplastic proteins (See Supplementary Data 5). ER endoplasmic reticulum. PM plasma membrane. **b** Volcano plots of 1156 proteins found in both the CL and APF showing their relative abundances based on their MS1 ion intensities. Green and orange points identify chloroplast and apoplast proteins, respectively. All others are colored in grey. The numbers of proteins with significant or insignificant differences in abundance between CL and APF are indicated in the corners of the graphs. The dashed lines indicate the significance boundaries based on both Log$_2$ FC ≥1 or ≥-1 and *P*-value of significance <0.05. **c** Functional analysis of the CL (2921 total), APF (1375 total), and APF-specific proteins (219 total) as determined by GO. Shown are the top 5-6 GO terms for each set based on the Cellular Compartment, Biological Process, and Molecular Function categories. Top left, Venn diagram showing the overlap of the two fractions.

GO terms for the 219 APF-specific proteins (Fig. 2c). Here, terms associated with the apoplast, extracellular space, and the cell wall dominated the Cellular Compartment GO assignments, whereas terms typically ascribed to the apoplast were seen for the Biological Process and Metabolic Functions categories, including carbohydrate metabolism, cell wall biosynthesis, polysaccharide metabolism, extracellular encapsulation, and chitinase and aspartyl esterase activities (Fig. 2c).

When the APF list was queried for specific proteins, we also saw a strong enrichment for those previously connected to the *Arabidopsis* apoplast[5,11]. In fact, the top nine and 54 of the top 90 proteins (60%) based on combined MS1 ion intensities were previously assigned to the apoplast, with many associated with carbohydrate metabolism/modification, proteases/peptidases, apoplastic enzymes, and activities linked to pathogen defense (Fig. 2c; Table 1). Included were various

glycosyl hydrolases, chitinases, xylanases, and β-galactosidases (BGAL), along with six subtilisin-type proteases (Supplementary Data 2).

Intriguingly, many apoplast proteins had previous connections to plant defense, including 33 of the top 50 (66%) cataloged as apoplastic in Table 1. For example, we identified seven out of the 17 BGAL isoforms encoded by *Arabidopsis* (family 35 glycosyl hydrolases) in the APF, some of which were previously implicated in releasing the flg22 MAMP from bacterial flagellins through removal of the decorating *O*-glucan moieties[15]. Also included were the SBT1.7 and SBT5.2 subtilases recently tied to the proteolytic release of flg22[19], with SBT1.7 being one of the most abundant apoplast constituents (Table 1; Supplementary Data 2). Similarly abundant APF proteins were the salicylic acid-responsive proteins Pathogenesis Related-5 (PR5), the PR2 β1-3

## Table 1 | Top Apoplastic Proteins in the Arabidopsis Leaf APF[a]

| Chromosome Location | Protein Name | Function/Activity | Location[b] | Rank[c] | Def.[d] |
|---|---|---|---|---|---|
| At1g75040 | PR5 | Pathogenesis Related-5/Thaumatin-like | Apo | 1 | + |
| At3g57260 | BGL2/PR2 | β-1-3 glucanase-2/Pathogenesis Related-2 | Apo | 2 | + |
| At3g55260 | HEX2 | β-hexosaminidase-1 | Apo/Vac | 3 | |
| At3g57240 | BG3 | β-1-2-glucanase-3 (family-17) | Apo | 4 | + |
| At3g08030 | | Cell wall DUF642 | CW | 5 | |
| At2g43570 | CHI | Chitinase | Apo | 6 | + |
| At1g09750 | | Aspartyl protease | Apo | 7 | |
| At2g14610 | PR1 | Pathogenesis Related-1 | Apo/CW | 8 | + |
| At5g10760 | AED1 | EDS1-depend-1 aspartyl protease | Apo | 9 | + |
| At5g26000 | BGLU38/TGG1 | β-galactosidase-38 | Apo/Chlo | 11 | + |
| At1g76160 | SKU5-like | Cu-oxidoreductase | Apo | 12 | + |
| At2g43590 | PR3-like | Pathogen Related-3 chitinase | Apo | 14 | + |
| At5g67360 | SBT1.7 | Subtilisin-like Ser protease | Apo | 15 | + |
| At4g23170 | CRK9/EP1 | Cys-rich receptor kinase-9 | Apo/PM | 16 | + |
| At5g08380 | GAL1 | α-galactosidase-1 | Apo/CW | 18 | |
| At5g17920 | MTS1 | Cobalamin-independent Methionine synthase | Apo/Cyto | 21 | |
| At1g21670 | | DPP6 domain containing | Apo/CW | 22 | |
| At3g01500 | BCA1/SABP3 | β-carbonic anhydrase | Apo/Chlo | 24 | |
| At3g18490 | ASPG1 | Aspartyl protease | Apo | 25 | + |
| At3g52840 | βGAL2 | β-galactosidase-2 | Apo | 27 | + |
| At4g20840 | BBE21 | Oligogalacturonide oxidase-2 | Apo | 28 | + |
| At1g79720 | | Aspartyl protease | Apo | 31 | |
| At2g28470 | βGAL8 | β-galactosidase-8 | Apo/CW | 32 | |
| At5g55450 | LTP4 | Bifunctional inhibitor/lipid transferase | Apo | 33 | |
| At1g29660 | GGL5 | Esterase/acyl transferase/lipase | Apo/CW | 35 | + |
| At5g64570 | XYL4 | β-D-xylosidase (family3) | Apo/CW | 36 | + |
| At4g27520 | ENODL2 | Early nodulin-like-2 | Apo | 37 | + |
| At2g10940 | | Bifunctional inhibitor/lipid transferase | Apo | 38 | |
| At3g14415 | GOX2 | Glycolate oxidase-2 | Apo/Per | 45 | + |
| At3g14210 | EMS1 | Glucosinolate hydrolase | Apo | 49 | + |
| At5g10560 | BXL6 | β-xylosidase-6 | Apo/Cyto | 50 | + |
| At2g46930 | PAE3 | Pectin acetyltransferase | Apo | 52 | + |
| At5g13690 | CYL1 | α-N-acetylglucosaminidase | Apo/Vac | 53 | |
| At5g13980 | | α-mannosidase (family 38) | Apo/CW | 54 | |
| At1g78830 | MNB1 | Curulin (mannose-binding) lectin | Apo/Golgi | 55 | |
| At2g38540 | LTP1 | Lipid transfer protein-1 | Apo/CW | 56 | |
| At5g25980 | BGLU37/TGG2 | β-glucosidase-2 | Apo/Vac | 58 | + |
| At5g11720 | AGLU1 | α-glucosidase (family 31) | Apo | 60 | |
| At3g07390 | AIR1 | Auxin-Induced in Root Culture-12 | Apo/PM | 61 | |
| At2g45470 | AGP8/FAC8 | Arabinogalactan Protein-8 | Apo/PM | 64 | |
| At1g65930 | clCDH | NADP-dep. Isocitrate dehydrogenase | Apo/Cyto | 65 | + |
| At5g20630 | GER3 | Germin-like protein-3 | Apo | 66 | |
| At5g47550 | CYS5 | Cystatin proteinase inhibitor-5 | Apo | 67 | + |
| At1g19570 | DHAR1 | Dehydroascorbate reductase-5 | Apo | 69 | + |
| At3g14420 | GOX1 | Glycolate oxidase-1 | Apo/Per | 71 | + |
| At2g36530 | ENO3 | Enolase-2 | Apo/Cyto | 72 | + |
| At4g12910 | SCPL20 | Ser-carboxypeptidase-like-20 | Apo | 77 | + |
| At4g23670 | MLP6 | Major Latex Protein-6 | Apo/Vac | 81 | + |
| At1g26380 | BBE3/FOX1 | FAD-linked oxidoreductase | CW/Cyto | 82 | + |
| At1g66970 | GDPDL1 | Glyceropholphosphodiesterase-like-1 | Apo | 83 | |
| At4g37800 | XTH7 | Xyloglucan endotransglucosylase | Apo | 86 | |
| At2g38010 | CER | Ceramidase-2 | Apo | 87 | + |
| At3g05730 | DEFL205 | Defensin-like protein 205 | Apo | 89 | + |
| At4g12880 | ENODL19 | Early Nodulin-Like Protein 19 | Apo | 90 | + |

[a]Top 50 ranked apoplast proteins identified in the APF based on MS1 ion counts as determined by LC-MS/MS.

[b]Most likely location(s) as defined in TAIR. Apo apoplast, Chlo chloroplast, CW cell wall, Cyto cytosol, Per peroxisome, PM plasma membrane, Vac vacuole.

[c]Abundance rank based on MS1 ion counts after removing Rubisco large and small subunits.

[d]+, Linked previously to biotic defense.

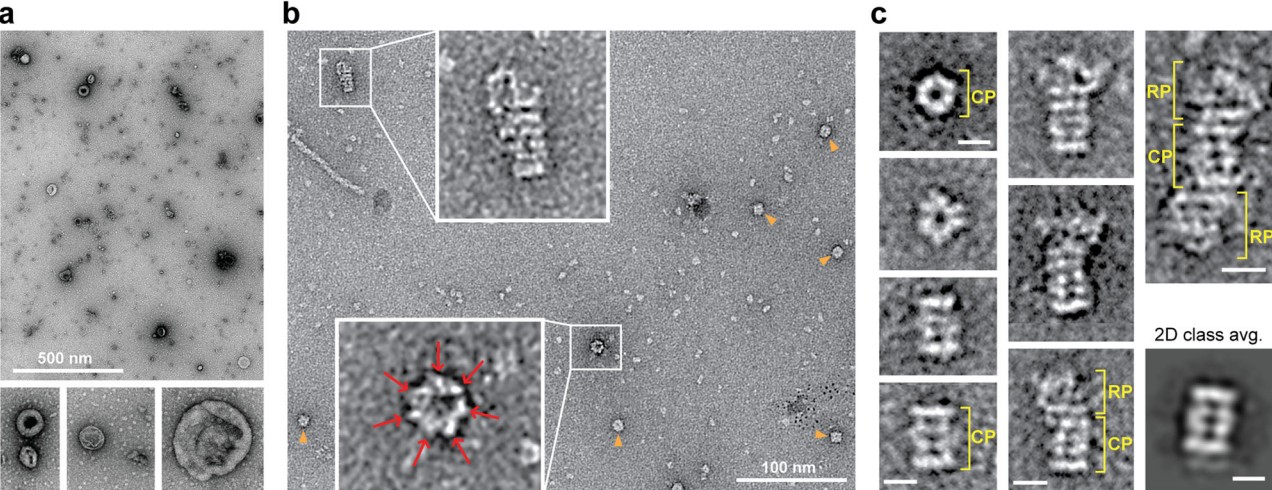

**Fig. 3 | Morphological detection of *Arabidopsis* ex-proteasomes by transmission electron microscopy (TEM) of the APFp.** The resuspended APFp was visualized by negative-stain TEM after enrichment from wild-type leaves**. a** A micrograph at low magnification showing EVs[7]. Bottom panels show close-up views of representative EVs. **b** A higher magnification view of the APFp identifying particles with the characteristic CP and RP-CP 26S proteasome architectures. The boxes highlight an end-on view of a CP barrel (arrows identifying the signature heptameric α-subunit ring) with its axial pore, and a side view of a CP singly capped with an RP. Orange arrowheads locate other possible top views of CP barrels. **c** Representative top and side views of individual CP particles with their 4-tiered αββα-subunit barrel assembly, side views of singly-capped CP-RP particles, and a rare doubly-capped RP-CP-RP 26S particle. The bottom right image shows a 2D class average for the side view of the CP barrel generated from 40 EM images. Scales bars = 8 nm.

glucanase0, several chitinases (CHI, PR1 and PR3-like) with critical roles in fungal defense, and several other factors involved in elicitor, oligosaccharide, ROS, and myrosinase/glucosinolate microbial and insect protection systems, including the HOPW1-INDUCED GENE1 protein that interacts with the *P. syringae* effector HOPW1-1 (Table 1; Supplementary Data 2 and 3**)**. Altogether, the proteomic data support a strong enrichment for the *Arabidopsis* apoplast using our isolation protocol, strengthen the connections between the apoplast and biotic defense, and highlight challenges inherent to the proteomic analyses of photosynthetic tissues.

## Morphological detection of proteasomes in the APF by TEM

Given the strong enrichment of known apoplast constituents, we searched our APF preparations for possible ex-proteasomes by negative-stain TEM based on the distinctive architecture of the CP and 26S complexes. To improve our chances, we concentrated large particles by centrifugation at 100,000 X*g* in the presence of 1 mM ATP to strengthen CP/RP binding[35,36] (Fig. 1a). Subsequent LC-MS/MS analysis of the APFp followed by volcano plots and GO comparisons confirmed that this pellet remained highly enriched in apoplastic proteins with minimal intracellular contamination (Supplementary Fig. 2; Supplementary Table 1; Supplementary Data 6-8). As reported previously[7,8], this centrifugal concentration enriched for a heterogeneous collection of 50-200-nm diameter EVs (Fig. 3a), some of which have been implicated in plant defense. Interestingly, increased magnification of the fluid revealed structures reminiscent of CPs both in size (10-12 nm diameter by 15-25 nm height) and shape[29,41], which were evident not only by a 2D class average but also with top and side views of individual particles (Fig. 3b, c). Most telling was the four-ringed CP barrel sometimes sufficiently resolved to detect its signature heptameric α-subunit rings surrounding the axial pores (Fig. 3b, c). By comparison, another large particle possibly in the APFp might have been Rubisco; however its 16-subunit, more ellipsoid barrel of 8 large and 8 small subunits intimately entwined without obvious tiers[42] was not obvious in the preparations. Less frequently, we also detected structures in the APFp resembling the CP singly capped by the asymmetric RP, and in rare situations, we detected CPs doubly capped with two RPs[29,41] (Fig. 3b, c), whose assemblies were presumably enforced by ATP added to the extraction buffer.

## Proteomic and immunodetection of ex-proteasomes

We then revisited our MS/MS descriptions of the APF and APFp in search for CP and RP subunits. As predicted[35,43], isoforms for almost all of the 26S holoproteasome subunits were readily detected in the CL (31 of 33 subunits; Fig. 4a), with the exception of RPN13 and RPN14/SEM1 from the RP Lid which are notoriously difficult to identify by shotgun MS analyses of purified *Arabidopsis* samples[44]. Remarkably, a similar search of the APF/APFp proteomes detected a comparable set containing all 14 CP polypeptides that assemble the seven-subunit α-rings (PAA(α1) – PAG(α7)) and β-rings (PBA(β1) - PBG(β7)), including the β1, β2, and β5 polypeptides that house the protease catalytic sites (Fig. 4a). Many of the *Arabidopsis* CP subunits (11 of 14) are encoded by two paralogous genes with varied expression levels and slightly distinct amino acid sequences[35]. For the most part, detection preferences comparing one CP subunit paralog over the other in the APF samples conformed to that seen with the CL and in line with their more robust expression in leaf tissue[35,45]. A notable difference was the strong preference for the PAA2 and PAD1 isoforms in the APF and APFp samples based on presence/absence and volcano plots (Fig. 4a; Supplementary Fig. 1b), indicating that the apoplast might harbor a unique CP subtype.

By contrast, the repertoires of RP polypeptides detected in the APF and APFp were less covered with some subunits/isoforms of the RP Base and Lid missing (Fig. 4a). Besides lacking RP13 and RPN14/SEM1, the most conspicuous absence was RPN10 that functions as both a Ub-receptor and an adaptor for the autophagic clearance of proteasomes[30]. Its known ability to partition between particle-bound and free forms[46,47] implies that RPN10 could dissociate from the RP before secretion into the apoplast. Preference for the CP versus RP in the apoplast was also seen when comparing relative protein abundances as calculated by combined MS1 ion intensities. Whereas the collective levels of the CP subunits defined by MS1 ion intensities were comparably higher in the APF/APFp versus CL, those for RP subunits were collectively lower especially in the APFp (Fig. 4c).

We also detected proteasome subunits in the APFp by immunoblot analysis with a library of anti-CP and anti-RP subunit antibodies[43]. As compared to a dilution series of the CL included to describe the sensitivity of each antisera, we readily detected PAG1(α7), PBA1(β1), and PBF1(β6) from the CP, but with the exception of RPN5, we often failed to detect other RP subunits such as RPN12, RPT2, and RPT4 (Fig. 4b). Here,

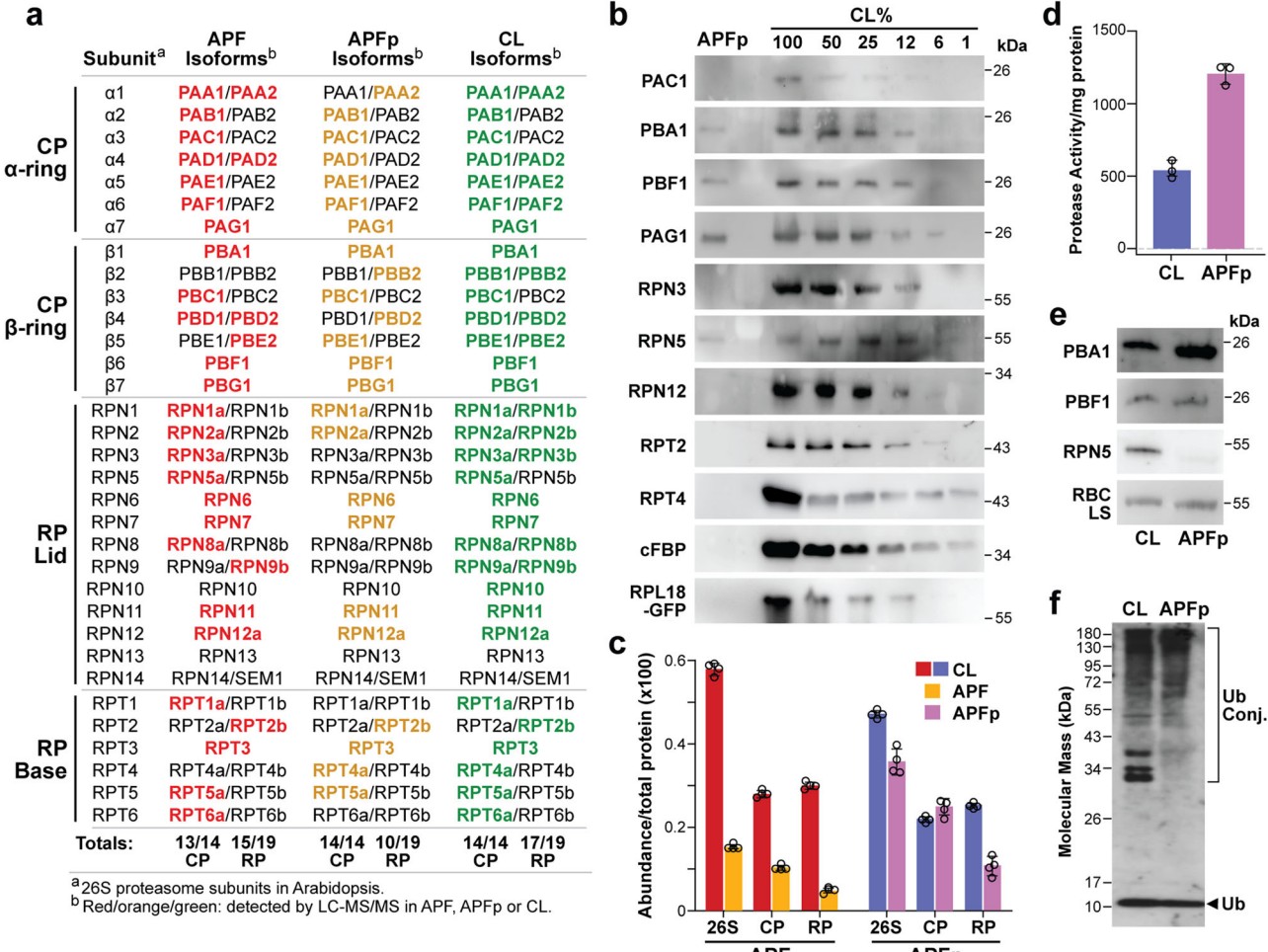

**Fig. 4 | Detection of ex-proteasomes in the apoplast by MS and immunoblot assays. a** Identification of proteasome subunits by LC-MS/MS analysis of the APF, APFp, and CL. Listed are all *Arabidopsis* proteasome subunit isoforms; those detected in the APF, APFp, and CL are marked in red, orange, and green type, respectively. The catalog was separated by the heptameric CP α and β rings, and the RP Lid and Base subcomplexes (see[35] for reference). Whereas most, if not all, of the 14 CP subunits were detected in the APF/APFp, a few of the 19 RP subunits were not (RPN10, RPN13, and RPN14/SEM1). **b** Immunoblot detection of ex-proteasome subunits in the APFp isolated from wild-type analyzed alongside a dilution series of the CL. Immunoblot analysis with anti-cFPB antibodies was included as a control. For a second control, the APFp was isolated from a line expressing a GFP-tagged subunit of the ribosome large-subunit protein RPL18 and immunoblotted with anti-GFP antibodies. **c** Relative abundance of the 26S proteasome and the CP and RP subcomplexes in the CL, APF, and APFp fractions as determined by MS/MS. The values were calculated by combining normalized MS1 data for all subunits of the indicated complexes. **d** Specific activity per mg protein for proteasomes in the APFp and CL based on hydrolysis of the CP substrate Suc-LLVY-AMC. **e** Immunoblotting for CP and RP subunits using equivalent amounts of proteolytic activity for the CL and APFp as determined in panel (**d**). **f** Immunoblot detection of Ub and Ub conjugates in the CL and APFp using anti-Ub antibodies. Ub conjugates and free Ub are located by the bracket and arrowhead, respectively. Bars in panels (**c**) and (**d**) reflect the mean (±SD) of four and three technical replicates, respectively. Individual data points are included.

low cytosolic contamination was confirmed by a failure to immunodetect cytosolic cFBP in the APFp from wild-type leaves, and the ribosome small subunit protein RPL18B in the APFp extracted from transgenic leaves expressing a GFP-tagged RPL18B fusion[48] (Fig. 4b).

For a third measure of proteasome enrichment, we quantified the specific activity of the proteasome in the APFp versus CL, based on hydrolysis of the fluorescent substrate succinyl-Leu-Leu-Val-Tyr-7-amino-4-methylcourmarin (Suc-LLVY-AMC) specifically designed for the chymotrypsin-like peptidase activity of the PBE1(β5) subunit[43,49]. Given the architecture of the CP, this activity can only be seen with intact particles and not the free β5 subunit[49]. As shown in Fig. 4d, e, the CP was enriched in the APFp based on both specific activity and subunit abundance as judged by immunoblot analysis with anti-CP and RP subunit antibodies. Collectively, the data agreed that the CP is the dominant ex-proteasome in the *Arabidopsis* apoplast with a smattering of capped 26S complexes also present. Interestingly, we also detected ubiquitylated proteins in the apoplast. As evident in Fig. 4f, the

characteristic smear of high molecular mass conjugates and possibly free Ub at ~10 kDa were seen by immunoblot analysis of both the CL and the APFp with anti-Ub antibodies.

## Proteasome inhibitor sensitivities in the APF matches that of the CP

A fourth approach to identify ex-proteasomes was through their sensitivity to proteasome-specific inhibitors. As shown in Fig. 5a, Suc-LLVY-AMC breakdown in the CL was strongly suppressed by the well-characterized CP inhibitors bortezomib (BTZ), MG132, and epoxomicin (Epo)[49], but not by an array of general peptidase/protease inhibitors, including bestatin, pepstatin, phenylmethylsulfonyl fluoride (PMSF), leupeptin, and E64 that block aminopeptidases, and aspartyl-, serine-, aspartyl/serine-, and cysteine-proteases, respectively, as well as to a plant protease inhibitor cocktail (PIC, Sigma-Aldrich) containing an inhibitor mix against serine-, cysteine-, amino-, and acid peptidases[15] (Fig. 5a). Comparable cleavage assays for the APF detected the identical

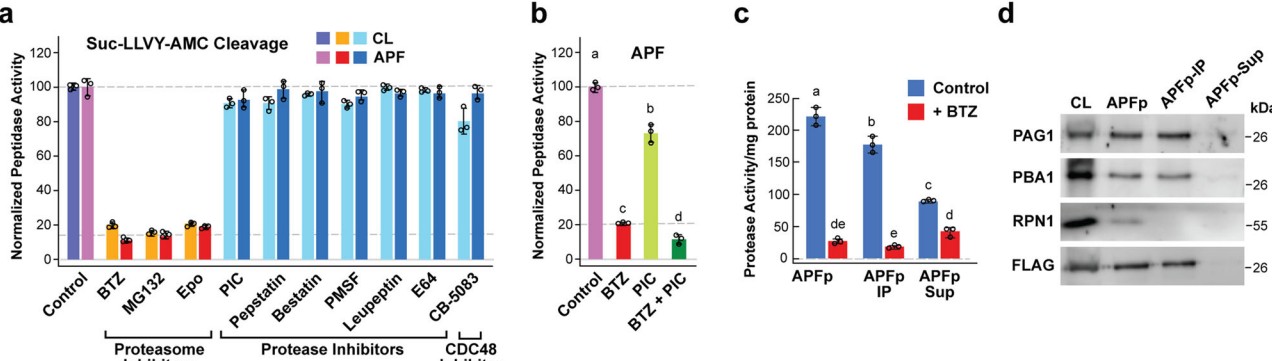

**Fig. 5 | Activity and inhibitor sensitivity of *Arabidopsis* ex-proteasomes.** Proteasome activity was assayed using the fluorogenic substrate Suc-LLVY-AMC. **a** Sensitivity of proteasomes in the CL and APF from *Arabidopsis* leaves to a collection of inhibitors designed to block either proteasomes (50 μM bortezomib (BTZ), 50 μM MG132, and 10 μM epoxomicin (Epo)), or various classes of peptidases/proteases (10 μM peptstatin, 15 μM bestatin, 100 μM PMSF, 10 μM leupeptin, and 10 μM E64), a 1X concentration of protease-inhibitor cocktail (PIC) that inhibits a collection of peptidases/proteases, or the activity of the CDC48 chaperone (10 μM CB-5083). The CL and APF were pretreated with the indicated concentrations for 5 min before assay. **b** Sensitivity of Suc-LLVY-AMC cleavage by the APF to a combination of 1X PIC and 50 μM BTZ. **c** and **d**, Association of Suc-LLVY-AMC cleavage with ex-proteasomes as judged by immune-depletion assays using a transgenic *Arabidopsis* line where the CP α7 subunit PAG1 was replaced with a FLAG-tagged variant. A resuspended APFp from *PAG1-FLAG pag1-1* leaves was depleted of proteasomes by immunoprecipitation with anti-FLAG beads. The APFp and the immunoprecipitated (IP) and supernatant (Sup) fractions were assayed for (**c**) protease activity using Suc-LLVY-AMC or for (**d**) proteasome subunits by immunoblotting with antibodies against CP (PAG1 and PBA1) and RP (RPN1) subunits or the FLAG epitope. Bars in panels a-c reflect the mean (±SD) of three technical replicates. Different letters above the bars indicate a significant difference from others using a one-way ANOVA followed by Tukey's test to determine significance. Individual data points are included.

array of inhibitor sensitivities and insensitivities confirming that the APF not only contains proteasome subunits but also catalytically-active CP complexes (Fig. 5a). Interestingly, PIC used together with BTZ effectively blocked residual APF activity against Suc-LLVY-AMC beyond that seem for BTZ alone (Fig. 5b), suggesting that a combination of proteasome and general protease inhibitors are most potent in protecting this substrate from APF-mediated proteolysis. As the CDC48 chaperone often functionally associates with the CP, we also tested its involvement in APF proteolysis using the selective inhibitor CB-5083[50]; no effect on Suc-LLVY-AMC cleavage activity was seen (Fig. 5a).

It has been reported that the peptidase activity of the CP can be enhanced by low concentrations of SDS (0.02%) presumably by opening the axial pores for substrate entry[51]. We did not observe such stimulation with the leaf APF (Supplementary Fig. 3). Nor did we observe an impact of ATP on ex-proteasome activity (Supplementary Fig. 3), implying that the CP and not the full 26S particle was mostly measured by the Suc-LLVY-AMC assays[43].

To further confirm that Suc-LLVY-AMC cleavage by the APF measured assembled CPs specifically and not individual β1, β2, and/or β5 subunits or other proteases, we immune-depleted proteasomes from the *Arabidopsis* APFp isolated from a transgenic *Arabidopsis* line where the non-catalytic PAG1(α7) CP subunit was genetically replaced with a functional FLAG-tagged variant[35]. As shown in Fig. 5c, d, anti-FLAG beads significantly removed both the CP peptidase activity and the CP subunits PAG1(α7) and PBA1(β1) and the RP subunit RPN1 from the APFp, with the ex-proteasome activity, CP subunits, and FLAG now enriched in the bound fraction. One additional concern was that the CP activity assays are typically done at neutral pH, whereas the plant apoplast is more acidic (pH ~4.7-5[10]), leading us to question whether proteasomes could be effective in this environment. However, when tested at various pHs, we found that apoplast proteasomes, like those in the CL, retained their BTZ-sensitive activity against Suc-LLVY-AMC over a broad pH range with robust cleavage even seen at pH 4.0 (Supplementary Fig. 4).

## Ex-proteasomes contribute to MAMP-mediated immune signaling

The existence of ex-proteasomes raised numerous questions as to their function(s) *in planta*. Given the link between extracellular

proteolysis and MAMP-mediated PTI defense[5,15,19], one intriguing possibility was a role in digesting pathogen proteins into immune signals that ultimately trigger protective ROS bursts, an example of which is the potent immunogenic flg22 peptide derived from bacterial flagellin (QRLSTGSRINSAKDDAAGLQIA from *P. syringae*). Accordingly, recent studies demonstrated that flg22 production requires not only apoplast β-glycosidases to remove the *O*-glycan moieties decorating flagellin but also apoplast protease(s) that release the flg22 peptide[15], of which the SBT5.2 and SBT1.7 subtilases were recently implicated[19].

Attempting to connect ex-proteasomes to MAMP signaling, we examined ROS bursts elicited in *Arabidopsis* leaf discs either by flg22 alone or by virulent *P. syringae* pv. *tomato* DC3000 cells after incubation in the APF, using a luminol/horseradish peroxidase cocktail to quantify ROS by fluorescence (Fig. 6a). Similar to prior studies[15,19], we extracted the APF with water, which we then amended with 50 μM ATP to maintain 26S proteasome stability. As expected, 100 nM of synthetic flg22 induced a strong ROS burst soon after mixing with the discs followed by a slow loss of fluorescence back to baseline over the next hr (Fig. 6b). Consistent with flg22 being downstream of ex-proteasomes, this burst was mostly insensitive to BTZ (Fig. 6b, c). We next measured the ROS burst from leaf discs incubated with *P. syringae* cells pretreated for 4 hr with the APF along-side the APF and cells tested alone. A burst comparable to flg22 was seen for the *P. syringae* + APF samples, while no burst was seen with the APF alone and only a modest burst was seen with bacterial cells alone (Fig. 6d). However, unlike flg22, the effect of the *P. syringae* + APF samples was strongly suppressed by preincubating the APF with BTZ prior to bacteria addition (Fig. 6d), thus plausibly linking ex-proteasomes to flg22 production.

To better illustrate this link quantitatively, we developed a measure for ROS burst in which the fluorescence signal measured with 50 μM ATP alone (Mock) every 2 min over a burst timeline (measured from min 2 to min 40) was subtracted from the actual fluorescence, and then the 20 adjusted values were integrated to generate a total luminescence intensity (TLI) value that reflected the cumulative strength of the burst. As shown in Fig. 6c, treating the leaf discs with flg22 alone induced a strong TLI signal, which was mostly retained upon pretreating the discs with BTZ. A similar robust TLI signal was seen when using *P. syringae* cells preincubated for 4 hr in the APF. But

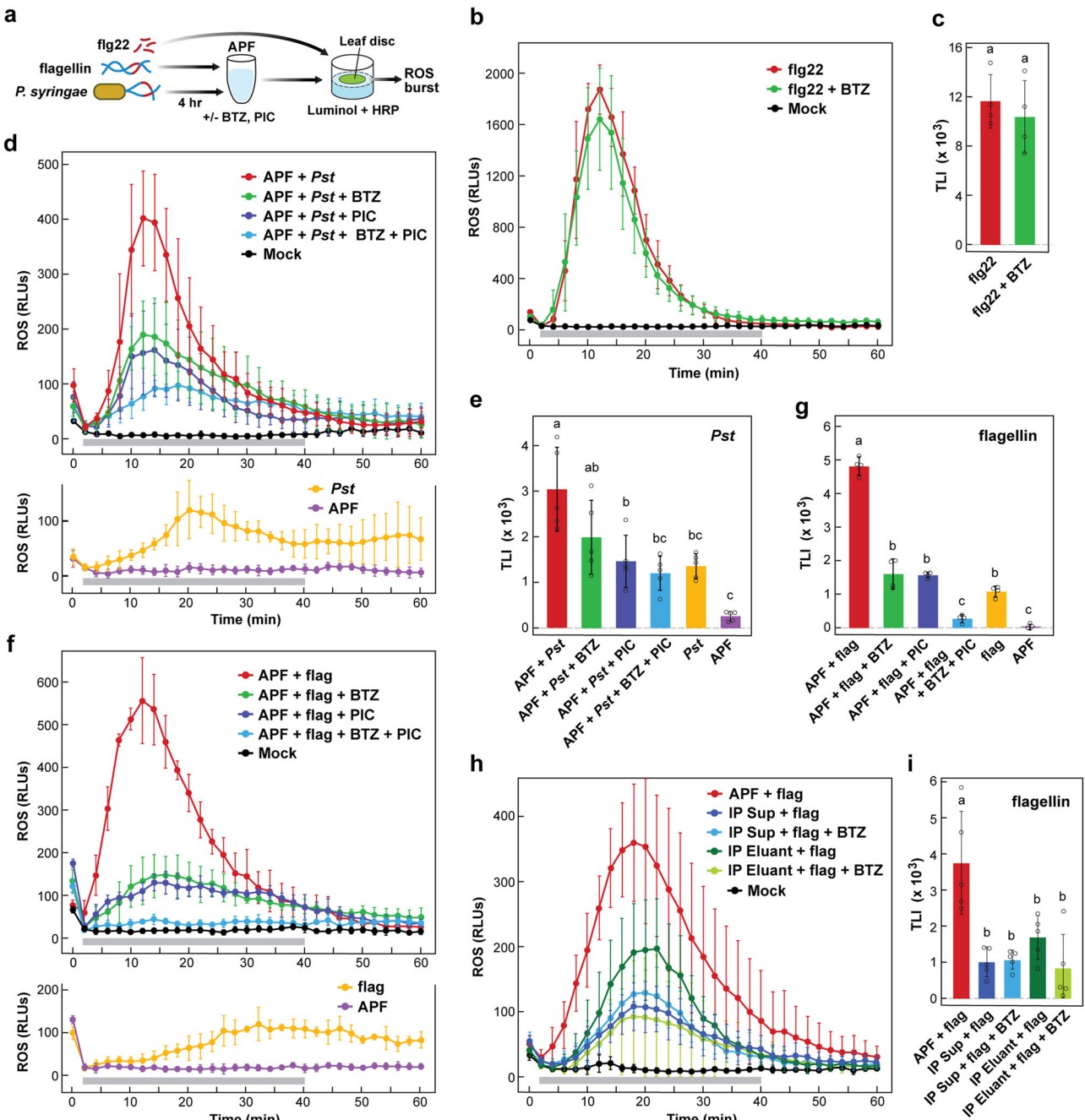

**Fig. 6 | The proteolytic activity of ex-proteasomes contributes to the ROS burst induced by the flg22 MAMP defense system in *Arabidopsis*. a** Protocol for observing the effects of ex-proteasomes on the ROS burst triggered by either flg22, *P. syringae* cells (*Pst*), or *P. syringae* flagellin. ROS was measured by relative fluorescence units (RLUs) every 2 min for leaf discs floating on a luminol/horseradish peroxidase (HRP) solution. Effect of 100 nmoles flg22 was tested by direct addition to the discs pretreated for 20 min with or without 62.5 μM BTZ. For *Pst* cells and flagellin, the cells/protein were incubated for 4 hr at room temperature in APF pretreated for 20 min without or with 62.5 μM BTZ or a 1X concentration of PIC before addition to the discs. **b** Effect of BTZ on the ROS burst elicited by flg22. **c**, Quantification of the ROS bursts measured in panel (b) by total luminescence intensity (TLI). **d** Effects of BTZ and PIC alone or in combination on the ROS burst elicited by APF-digested *Pst* cells. The lower panel shows the effects of *Pst* and APF alone. **e** Quantification of the ROS bursts measured in panel (d) by TLI. **f** Effects of BTZ and PIC alone or in combination on the ROS burst elicited by APF-digested flagellin. The lower panel shows the effects of flagellin and APF alone.

**g**, Quantification of the ROS bursts measured in panel (**f**) by TLI. **h** Immune-depletion of ex-proteasomes from the APF dampens the ROS burst elicited by flagellin. The APF isolated from *PAG1-FLAG pag1-1* leaves were depleted of ex-proteasomes with anti-FLAG beads and the bound proteasomes were then released with FLAG peptide. Equal volumes of APF and the supernatant (Sup) and Eluant fractions (IP) were pretreated ±62.5 μM BTZ, and then incubated with flagellin for 4 hr before adding to the discs. **i** Quantification of the ROS bursts in panel (**h**) by TLI. Each point in panels (**b**), (**c**), (**f**), and (**h**) represents the mean of four, five, four, and five biological replicates (±SD), respectively. The horizontal grey bars indicate the time frames for measuring TLI. TLI values were calculated by subtracting the RLU values generated with the Mock from those generated with the test sample at each time point and integrating the adjusted RLU values to generate the TLI (±SD). The individual data points are included. Letters indicate a significant difference from others using a one-way ANOVA followed by Tukey's test to determine significance. Mock, ROS measured with APF extraction buffer (50 μM ATP) alone.

in this case, the TLI response was suppressed ~35% after pretreating the APF with BTZ before adding the cells (Fig. 6e).

Given that the ROS burst induced by *P. syringae* cells could have been triggered by MAMPs released from a variety of bacterial components and not just flagellin, we tested APF mixed with purified flagellin alone, which was enriched from *P. syringae* pv *tomato* DC3000 by a combination of mechanical stress, acid denaturation, and clarification[52] (Supplementary Fig. 5a). Initial studies demonstrated that most of the 31-kDa flagellin polypeptide was degraded by the APF after a 4-hr incubation at room temperature (Supplementary Fig. 5b), consistent with prior studies showing that the APF has activit(ies) that can break down this glycoprotein[15,19]. As demonstrated in Fig. 6f, g, APF-digested flagellin induced a robust ROS burst and TLI response by itself, which was strongly suppressed by pretreating the APF with BTZ before flagellin addition, thus directly connecting flagellin specifically to the ROS burst elicited by ex-proteasomes.

Interestingly, we noticed both *P. syringae* cells and purified flagellin alone invariably stimulated modest but prolonged ROS responses and TLI values (~44% and 23% relative to those also with APF, respectively) when added to the leaf discs with fluorescence still evident after 1 hr (Fig. 6d, g). We presume that these delayed kinetics reflect the time needed for *P. syringae* cells and flagellin penetration into the leaf apoplast and subsequent cleavage into MAMPs such as flg22 by endogenous ex-proteasomes and/or other peptidases/proteases. Given the likelihood that other proteases such as SBT5.2 and SBT1.7[19] also participate in generating MAMPs from flagellin, we tested the general PIC mix by itself and together with BTZ. While PIC alone could modestly suppress the *P. syringae* cell- and flagellin-derived responses, the suppression was substantially stronger when combined with BTZ, implying that both ex-proteasomes and other APF proteases work in concert (Fig. 6d, g).

One complication to these assays is that extracellular ATP alone can trigger ROS bursts by acting as a tissue damage signal working through plasma membrane-bound purinoceptors[39,53]. Here, we tested this possibility by measuring the ROS response of leaf discs treated with APF in the presence or absence of ATP and flg22. Whereas no ROS burst and measurable TLI values were evident when APF alone or APF plus ATP up to 500 μM was tested, robust ROS burst and TLI values were seen when flg22 was also added if the ATP concentrations was kept at 50 μM or below (Supplementary Fig. 6).

To further confirm that BTZ suppresses flagellin-mediated ROS by inhibiting ex-proteasomes, we tested MG132 and Epo that work either as a reversible proteasome inhibitor similar to BTZ or as an irreversible inhibitor that covalently modifies the CP active sites, respectively[43,49] (see Fig. 5a). In side-by-side assays, increasing concentrations of each of the three inhibitors effectively blocked the flagellin-induced ROS burst with sub-micromolar IC50 values calculated based on TLI suppression (Fig. 7a–c). Notably, none of the inhibitors significantly blocked the ROS burst when added to leaf discs along with flg22 alone, indicating that the inhibitors blocked proteasomes working before flg22 perception (Supplementary Fig. 7). Inhibitor effectiveness (BTZ>Epo>MG132) in these relatively crude APF + flagellin ROS burst assays correlated remarkably well with their potency against purified *Arabidopsis* 26S proteasomes[43] and in bioassays on *Arabidopsis* seedling growth[54].

As final confirmation that ex-proteasomes within the APF generate MAMPs from flagellin to elicit a PTI response, we tested the efficacy of APF samples isolated from *PAG1-FLAG pag1-1* leaves which were immune-depleted of ex-proteasomes with anti-FLAG antibody beads. While a robust ROS burst was induced by flagellin after preincubation with the *PAG1-FLAG pag1-1* APF, this burst was substantially diminished (~4 fold) after immune-depletion (Fig. 6h, i). Most of the residual activity in the supernatant was insensitive to BTZ, indicating that it involved other peptidases/proteases besides ex-proteasomes. By contrast, much of the depleted activity could be, at least partially,

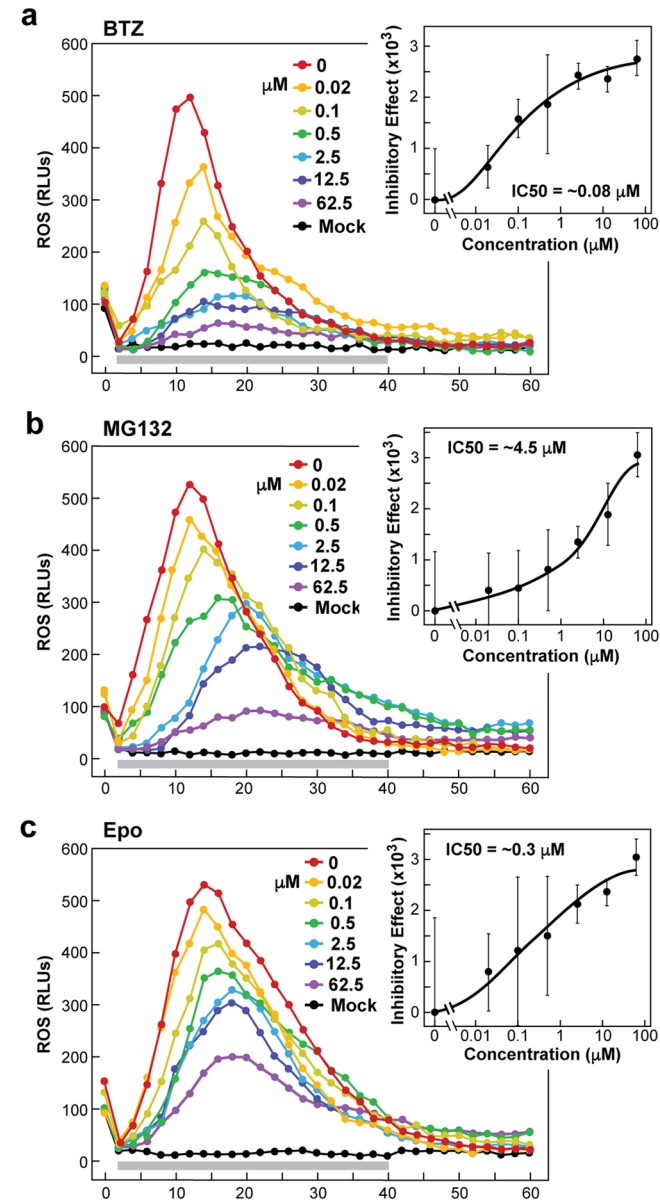

**Fig. 7 | Efficacy of proteasome inhibitors in suppressing the ROS burst elicited by flagellin.** *Arabidopsis* APF was incubated with purified *P. syringae* flagellin for 4 hr at room temperature with or without increasing concentrations of BTZ (**a**), MG132 (**b**), or Epo (**c**), or an equivalent volume of DMSO (Mock), and then added to leaf discs floating on a luminol/horseradish peroxidase solution. ROS bursts were assayed every 2 min by relative fluorescence units (RLUs). n = four biological replicates (±SD). Horizontal grey bars indicate the measurement time frames for determining the TLI (min 2 to 40). The right panels show the effects of increasing concentrations of inhibitor on fluorescence output (*n* = 4 biological replicates). The inhibitory effect was calculated first by subtracting the mean of the Mock TLI from the individual TLI values for each of the treated samples. Dose-response curves for the inhibitors were generated by subtracting mean of the TLI without inhibitor (0 μM) from the mean TLI value at each inhibitor concentration (±SD). IC50 values were estimated from the dose-response curves. Each inhibitory effect data point represents the mean of four biological replicates (±SD).

recovered by releasing ex-proteasomes from the beads with the FLAG peptide (Eluant) followed by incubating the eluant with flagellin (Fig. 6h, i). This eluant activity was partially sensitive to BTZ confirming ex-proteasome action. Taken together, the inhibitor and immune-depletion studies demonstrate that the MAMP-derived ROS bursts from flagellin involve both ex-proteasomes and other apoplastic

peptidases/proteases working in concert to generate effective immunogenic fragments.

## The *P. syringae* proteasome inhibitor syringolin A (SylA) blocks flagellin-mediated ROS bursts

An intriguing facet of *P. syringae* pathology is the synthesis and secretion of the non-ribosomal peptide SylA (Fig. 8a) by certain virulent strains such as pv *tomato* DC3000 to enhance colonization and infectivity[25,26]. Prior observations that SylA is a potent proteasome inhibitor through irreversible inactivation of the caspase-like, trypsin-like, and chymotrypsin-like activities provided by the β1, β2, and 5 subunits, respectively[55], led us to surmise that one function of this toxin is to suppress MAMP-mediated PTI by blocking ex-proteasome activity. Here, we confirmed this by first showing that micromolar concentrations of SylA, like BTZ, MG132 and Epo used above, not only effectively inhibit the proteolytic activity of affinity-purified *Arabidopsis* proteasomes and those in the CL, but also those of ex-proteasomes in the APF based on Suc-LLVY-AMC peptidase assays (Fig. 8b). Strikingly, SylA also inhibited the MAMP-mediated ROS burst triggered in leaf discs by APF-digested flagellin with an IC50 of approximately 0.5 μM as measured by TLI values (Fig. 8e), presumably by slowing conversion of flagellin into flg22 and other MAMPs. By contrast, SylA, like Epo, had no significant effect on the ROS burst elicited by flg22 alone, again illustrating that ex-proteasomes work upstream of this MAMP (Fig. 8c, d).

## Mechanism for ex-proteasome secretion unrelated to autophagy

Our discovery of ex-proteasomes raised the intriguing question as to their export mechanism(s). Given the role for autophagy in shuttling impaired proteasomes to the vacuole for clearance[56], the vesicular secretion of microbial defense RNAs into the apoplast[7,8], and a recent connection between defense-triggered cell wall lignification and autophagic transport to the apoplast[57], one obvious possibility was an autophagy-type route that encapsulates proteasomes into either cytoplasm autophagosomes formed de novo or related amphisomes arising from multivesicular bodies. To test these scenarios, we assessed ex-proteasome levels in the APFp extracted from mutants either compromised in autophagosome assembly (*atg5-1*, *atg7-2*, and *atg12a-1 atg12b-2*[58]), defective in RPN10 that functions as both a Ub receptor and an autophagic receptor for proteasomes (*rpn10-1*[56]), or missing the cell death-related endosomal FYVE/SYLF (CSF)-1 protein needed to mature autophagosomes into amphisomes (*csf-1*[59]). We also examined the APFp from the *atg5-1 sid2* double mutant that has been reported to eliminate the impact of salicylic acid that hyperaccumulates specifically in *Arabidopsis atg5-1* plants[60]. Surprisingly, none of the mutations strongly influenced the protein profile of the APFp as seen by SDS-PAGE nor decreased its concentration of the CP subunits PAG1(α7) and PBA1(β1) as seen by immunoblotting (Supplementary Fig. 8a). The mutations also failed to dampen the levels of active APF proteasomes as judged by Suc-LLVY-AMC cleavage assays with or without BTZ (Supplementary Fig. 8b).

For a more in-depth analysis, we defined the protein composition of the APFp in the mutant collection by LC-MS/MS. As shown by volcano plots, the autophagy-related mutations had little influence on the APFp proteome generally and on the abundance of proteasomes specifically, using normalized values based on combined MS1 ion intensities for the comparisons (Supplementary Fig. 9). Only the *rpn10-1* mutant, which raises intracellular proteasome levels by blocking dysfunctional particles turnover[56], marginally elevated ex-proteasome levels in the APFp as compared to that in wild type (Supplementary Fig. 9). Consequently, we were left to conclude that another route outside of canonical autophagy is responsible for ex-proteasome secretion and that autophagy has little impact on the overall protein profile of the APF.

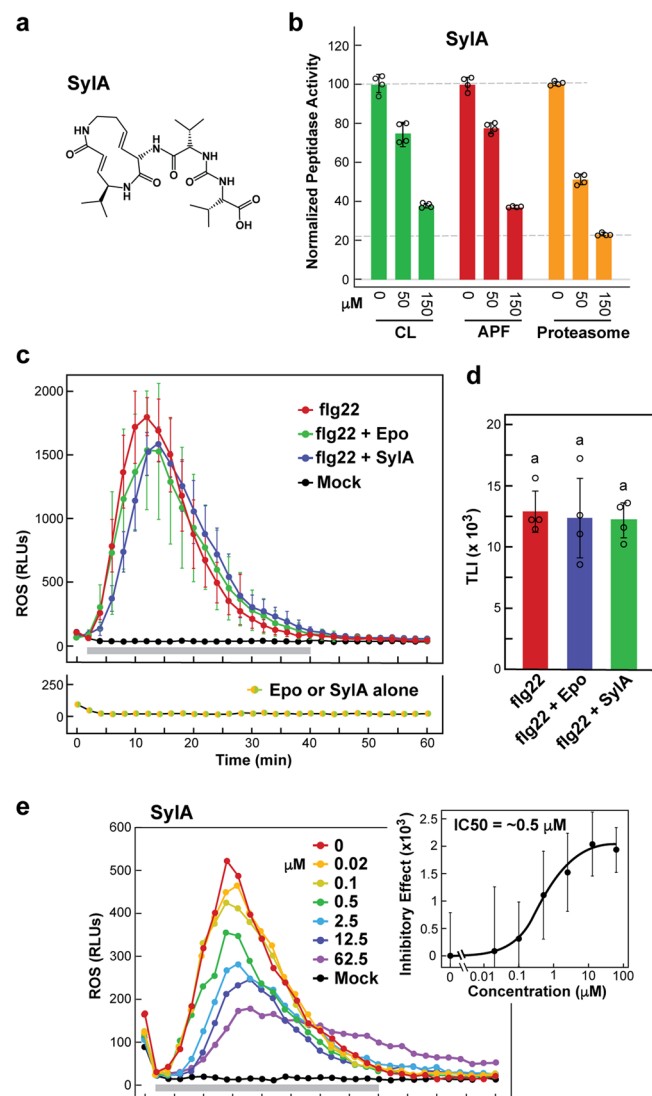

**Fig. 8 | The *P. syringae* toxin syringolin-A (SylA) inhibits ex-proteasome activity and suppresses ROS bursts triggered by flagellin but not flg22. a** Chemical structure of SylA. **b** *Arabidopsis* ex-proteasome activity is inhibited by SylA. The crude lysate (CL), APF, and affinity-purified proteasomes were extracted from *PAG1-FLAG pag1-1* leaves, pretreated for 20 min without or with 50 or 150 μM of SylA or an equivalent volume of DMSO (Mock), and assayed for proteasomes activity using the fluorogenic substrate Suc-LLVY-AMC. Bars reflect the mean (±SD) of four technical replicates with the values normalized to the reactions obtained without SylA. Individual data points are included. **c** SylA, like Epo, does not block the ROS burst elicited by flg22. Leaf discs were incubated with flg22 with or without a 10-min pretreatment with 62.5 μM of Epo or SylA, or an equivalent volume of DMSO, and assayed for ROS burst as in Fig. 6. The left panel show the time course for the mean RLUs measured with five biological replicates (±SD). Horizontal grey bar indicates the measurement timeframe for determining the total luminescence intensity (TLI). **d** Quantification of the ROS bursts measured by each treatment in panel (**c**) by TLI generated over minutes 2 to 40 (±SD). Individual data points are included. The letter a above the bars indicates no significant difference from others using a one-way ANOVA followed by Tukey's test to determine significance. **e** Addition of SylA to the APF suppresses the ROS burst elicited by flagellin. APF was incubated with purified *P. syringae* flagellin for 4 hr with or without a 10-min pretreatment with increasing concentrations of SylA, or an equivalent volume of DMSO, and assayed for ROS burst as in Fig. 6. The left panel shows the time course for the mean RLUs measured as in panel (c) with four biological replicates (±SD). The right panel shows the inhibitory effect of SylA calculated by an IC50 value as in Fig. 6a-c. Each point represents the mean of four biological replicates (±SD).

## Discussion

Despite its relevance to a number of physiological processes, the apoplast still remains one of the least understood plant compartments. Major challenges are: (i) the lack of boundaries as it encompasses all space outside of the plasma membrane including the porous cell wall, aqueous and gaseous regions, and xylem, (ii) its unavoidable loss of integrity upon tissue homogenization, and (iii) its dilute concentration of constituents. Using an oriented and low $g$ centrifugation strategy following mild vacuum infiltration, we extracted leaf APF with minimal cytosolic contamination, as judged by cytoplasmic markers (cFBP, actin, and ribosomes) and chlorophyll levels. LC-MS/MS of these preparations together with those concentrated by high-speed centrifugation (APFp) developed an enriched catalog of apoplast proteins that generally agreed with prior studies (e.g. refs. 5,11,) and predicted TAIR assignments (https://www.Arabidopsis.org), but was substantially expanded to include additional apoplast proteins not yet assigned as well as missing some non-apoplast proteins that might have been previously included by less stringent isolation methods. Nevertheless, we acknowledge that apoplastic assignments are inherently challenged by; (i) cell breakage, (ii) proteins with misassigned localizations or found in multiple compartments including the apoplast, (iii) proteins released *en masse* from other compartments by programmed cell death such as during xylogenesis, and/or (iv) proteins functioning in other compartments potentially being exported deliberately into the APF for breakdown. Rubisco, which is cleared by several routes[61], might be an example of the latter (see below).

Using stringent MS criteria, we developed a comprehensive catalog of apoplast residents, a substantial number of which were found specifically in this fluid versus clarified leaf lysates. The total list contains numerous proteases/peptidases, glycanases, and other hydrolytic enzymes potentially relevant to cell wall dynamics and pathogen defense[5,13]. As this collection of lytic activities should create a relatively hostile environment for normal metabolic functions, we imagine that one of the main APF functions is biotic protection. Defense-related constituents include PR5, the β1-3 glucanase PR2, several α- and β-glucosidases, cystatin, the CHI, PR1 and PR3-like chitinases, the β-glycosidase BGAL1 and subtilases SBT5.2 and SBT1.7 recently connected to MAMP-triggered immune signaling[15,16,19], and enzymes associated with myrosinate/glucosinolate synthesis and EVs possibly connected to RNAi-mediated apoplastic defense[7,8]. Accordingly, a majority of the 50 top-ranked apoplast proteins had previous links to biotic defense (Table 1, Supplementary Table 1). That said, we did not detect by MS several markers previously associated with EVs, including TET8, PEN1, and PEN3, but did find the pathogen-induced and secreted PTLA1 and PTLA2 lipases.

Nonetheless, our apoplast catalog, and likely those from other MS analyses of this fluid[4,9,12,14,27,28], remain contaminated with Rubisco and other chloroplast proteins, which skew enrichment and GO analysis of these preparations despite having little to no chlorophyll and few proteins from other compartments. Our strategy to better assess chloroplast contamination was to quantify proteins not by peptide spectral matches but via a semi-quantitative measure of protein abundance based on combined MS1 ion intensities (after removing Rubisco) that would minimize the bias generated by Rubisco and chloroplasts. While MS1 ion intensities (e-values) are well known to vary substantially among peptides/proteins, we hypothesized that such variations among samples would even out when aggregating the large protein/peptide numbers seen for our APF and CL preparations.

The only chloroplast proteins that remained dominant in the APF/APFp after this normalization strategy were the Rubisco large and small subunits, which were still prevalent in our apoplastic datasets despite near undetectable levels of chlorophyll (<1%). While the reason(s) behind this contamination are numerous, statistical correlations suggest that this 'apoplastic' Rubisco results from cell breakage. Other intriguing possibilities are that Rubisco is unavoidably released during programmed cell death and/or is deliberately exported into the apoplast as part of a homeostatic mechanism that clears dysfunctional complexes[61]. That we detected a faster migrating species of the Rubisco large subunit during SDS-PAGE of the APF could reflect the beginnings of such proteolysis.

Using a myriad of approaches, including TEM, MS, immunoblotting, immune-depletion, specific activity assays, and inhibitor sensitivities, we confirmed our initial speculation that the *Arabidopsis* apoplast contains functional proteasomes, mainly as free CPs but also as CP complexes capped with one or two RPs. The presence of both particles as seen by TEM was confirmed by the ability of both CP and RP subunits to co-sediment at 100,000 X$g$ with proteasome-specific activity (Fig. 4a), and copurify with anti-FLAG antibody beads using tagged proteasomes where only one non-catalytic subunit of the CP subcomplex harbored the FLAG epitope (Fig. 4d,e). The preferential accumulation of the CP was further supported by MS/MS analyses that detected all α and β subunits needed to assemble a functional CP but not all RP subunits (e.g., missing RPN10, RPN13 and RPN14/SEM1) and fewer isoforms (Fig. 4a). Our reanalysis of prior proteomic data came to the same conclusion for the apoplast from other plant species[4,9,12,14,27,28]. Coincidentally, the mammalian extracellular fluid also preferentially accumulates CPs[31].

The presence of ex-proteasomes raises intriguing questions as to their function(s). Possibilities include roles in: (i) protein homeostasis outside the cell, where a bevy of proteases and other hydrolytic enzymes are secreted, (ii) zymogen activation, (iii) regulating the half-life of secreted peptide hormones and other growth factors involved in intercellular signaling, and/or (iv) involvement in pathogen defense. Our studies with flagellin and its derivative flg22 provide a compelling connection to the host PTI system whereby plant ex-proteasomes aid in immune signaling by helping convert conserved bacterial proteins into MAMPs detected by host pattern recognition receptors such as FLS2. Here, we specifically showed by inhibitor and immune-depletion assays that ex-proteasomes in the APF can convert *P. syringae* flagellin into one or more MAMPs capable of triggering robust ROS signals central to basal defense. That the proteasome inhibitors failed to suppress flg22-triggered ROS but was effective for *P. syringae* cells and flagellin provides a strong argument that the inhibitors do not impact cytoplasmic proteasomes that might work downstream of flg22.

Strikingly, the ROS burst induced upon preincubating flagellin with the APF was inhibited by well-established proteasome inhibitors as well as by the SylA toxin secreted from virulent *P. syringae* strains to enhance infectivity. In fact, micromolar concentrations of SylA were not only effective in blocking *Arabidopsis* proteasome/ex-proteasome activity but also in attenuating ROS bursts triggered by the APF-mediated digestion of flagellin. Consequently, we add SylA to the arms race between *P. syringae* and its plant hosts, first as shown here by SylA inhibiting ex-proteasomes engaged in early PTI detection through MAMPs, and later as other studies have shown, by suppressing effector-triggered immunity mechanisms involving intracellular proteasomes working alone or in concert with the Ub conjugation system[62,63]. We note that additional links between ex-proteasomes in PTI signaling and pathogen counter responses are possible. As examples, the papaya ring spot virus HcPro protein has been reported to interact with and inhibit the proteolytic activity of the CP via binding to the PAA1(α1) subunit in an effort to enhance viral particle accumulation[64], while the fungal toxin higginsianin-B from *Colletotrichum higginsianum* suppresses defense signaling through host proteasome inhibition[65] possibly by a strategy similar to that of SylA.

At present, it remains unclear how ex-proteasomes help disassemble flagellins into MAMPs. As with previous studies[15,19], we found that purified flagellin can be rapidly broken down by the APF in vitro, which we also showed is accomplished by a BTZ- and PIC-sensitive process (Supplementary Fig. 5b). Unfortunately, MS analysis of the digests by us have failed so far to identify flg22-related peptides

generated solely by ex-proteasomes (data not shown), presumably because the apoplast is remarkably rich in a wide array of peptidases/ proteases (e.g., Table 1). In fact, it is likely that ex-proteasomes work in tandem with other apoplastic proteases such as the SBT5.2 and SBT1.7 subtilases[19] based on the need for both proteasome inhibitors and the general PIC mix to effectively block *P. syringae* cell- and flagellin-mediated ROS bursts. Similarly, SylA by itself also does not completely suppress flagellin-mediated ROS bursts mediated by the APF, indicating the need for other components besides ex-proteasomes.

It is conceivable that ex-proteasomes begin MAMP release followed by further trimming with other proteases. In fact, while cytosolic proteasomes typically generate peptides 4-11 amino acids in length which are further disassembled by other peptidases[29], there are known situations where proteasomes, inherently working as processive exo-peptidases, only partially break down the parent protein to release much larger biologically-active fragments. Examples include the partial processing of antigens into peptides accessible to the mammalian major histocompatibility complex (MHC) class I receptors, and the partial cleavage of "slippery" substrates such as of the mammalian NF-κB precursor p105, and the yeast transcriptional regulators Spt23 and Def1, whose structures impair processive unfolding and digestion[66]. Based on the 3D structure of flagellin[67], we image that the anti-parallel helix-transition-helix segment encompassing the flg22 sequence, along with decorating *O*-linked glycans, could stall ex-proteasome digestion around this site, thus leaving it intact for break down by other proteases. It is also likely that ex-proteasomes help generate other protein-based MAMPs besides flg22. An obvious possibility is the flgII-28 MAMP generated from a more-C-terminal conserved region of *P. syringae* flagellin, which is recognized by a distinct pattern recognition receptor FLS3 in several *Solanaceous* species[68].

Clearly, the discovery of ex-proteasomes raises a number of interesting questions related to their origin and possible regulation. Our initial speculation that an autophagy-type mechanism drives secretion was not supported by the proteomic analyses of the APFp extracted from various autophagy mutants. While our MS/MS analyses implied that ex-proteasomes are structurally equivalent to those inside *Arabidopsis* cells, it is conceivable that they interact with unique apoplastic regulators and/or bear novel post-translational modifications that aid export, activity, and/or substrate preferences. The lack of detectable RPN10 in the *Arabidopsis* leaf APF/APFp, which is needed to stabilize interactions between the Lid and Base subcomplexes of the RP[69], and the preferential accumulation of the PAA1(α1) and PAD1(α4) subunit isoforms in the CP raise the intriguing notion that plant ex-proteasomes are a unique proteolytic complex, possibly analogous to mammalian immunoproteasomes and thymoproteasomes that selectively incorporate unique CP β subunits for basal defense and antigen presentation[70]. Also intriguing is the possibility that ex-proteasome activity is enhanced at the sites of pathogen invasion to heighten MAMP synthesis.

Another interesting feature of the *Arabidopsis* apoplast preparations was our immunodetection of Ub and Ub conjugates, suggesting that the apoplast harbors a functional Ub/proteasome proteolytic system. Given that the apoplast contains surprisingly high concentrations of ATP[71] along with our MS detection of Ub-activating enzyme (UBA1; At2g30110; Supplementary Data 7), which initiates the Ub the -conjugation cascade, in the APFp, it is possible that these apoplast conjugates were assembled de novo as opposed to being constructed in the cytoplasm and then secreted. Notably, both free Ub and Ub conjugates have also been detected in mammalian extracellular fluids with possible roles in immune responses[72]. Despite the unanswered questions, our demonstration of ex-proteasomes and their connections to MAMP-triggered innate immunity should raise sufficient interest to warrant further studies of this extracellular proteolytic machine and its role(s) in plant apoplast biology and pathogen defense.

## Methods

### Plant materials and APF isolation

Wild-type and homozygous mutant plant lines were derived from the *A, thaliana* ecotype Col-0 background, and included the *atg5-1, atg7-2, atg12a-1 atg12b-1, rpn10-1, csf-1,* and *atg5-1 sid-2* mutants as previously described[56,58–60]. A transgenic line expressing a GFP-tagged version of the ribosome large subunit protein RPL18 was previously reported[48]. For the immune-depletion assays, the tagged proteasome line incorporating the PAG1(α7) CP subunit bearing a C-terminal FLAG tag complementing the lethal *pag1-1* background was used to enrich for proteasomes from the CL and APF[35].

For APF isolations, fully developed, undamaged leaves were selected from 5-6-week-old plants grown at 25 °C in a short-day photoperiod (8 hr-light/16-hr dark) and severed with a razor blade near the petiole/stem junction. Ten leaves as a batch were completely submerged in a 60-mL syringe containing 30–35 mL of APF extraction buffer, and infiltrated under a hand plunger-generated vacuum followed by a slow release to minimize cellular damage. For the biochemical analyses of the APF, the extraction buffer included 20 mM MES-HCl (pH, 6.0), 2 mM $CaCl_2$, 10 mM NaCl, and 0.05 or 1 mM ATP for APF and APFp preparations, respectively, while for the ROS burst assays it included only 50 μM ATP (pH ~6). See Fig. 1a for a diagrammatic explanation of the protocol. After removing damaged leaves, 10 leaves were bundled through their petioles using surgical tape, suspended at the top of the 50 mL syringe with their petioles up, and centrifuged at 600 X*g* for 20 min at 4 °C to collect the infiltrate into a nested 300-mL centrifuge tube. Large particles and cellular debris were removed by passing the infiltrate through a 0.2-μm Acrodisc syringe filter (Pall Corp.). Typically, 10 batches of 10 leaves yielded ~2 mL of clarified APF. This APF was either used directly, concentrated by lyophilization for MS analysis, or enriched for larger particles by further centrifugation at 100,000 X *g* for 120 min at 4 °C. The resulting APFp from 4 mL of fluid was resuspended in 50-100 μL of APF extraction buffer before use. Both the APF and resuspended APFp were stored at -80 °C. Chloroplast contamination was assayed by extracting the samples in 80% ethanol and followed by spectrophotometric quantification of chlorophyll at 600 nm.

### Transmission electron microscopy (TEM) after negative staining

Three-μL aliquots of the resuspended APFp were absorbed for 60 sec onto carbon-coated 200 mesh copper grids (01840-F, Ted Pella, Redding, CA), which had been glow discharged for 30 sec in a Solarus 950 plasma cleaner (Gatan, Peasanton, CA). After incubation, the grids were washed five times with ultrapure water and stained with freshly prepared 0.75% (w/v) uranyl formate for 2 min. Excess uranyl formate was blotted off with filter paper (Whatman No.2, Fisher Scientific) before air drying. Samples were imaged with a JEOL JEM-1400Plus TEM microscope (JEOL USA) at an operating voltage of 120 kV with a NanoSprint15-MkII 16-megapixel sCMOS camera (Advanced Microscopy Techniques) at 120 kV. Micrographs were collected as tiff files at a nominal magnification of 30,000x and a pixel size 3.54 Å/pixel. The micrographs were first converted to mrc files with inverted contrast using the e2proc2d.py script from EMAN2[73]. 2D-class-average side views of the CP barrel were generated in CryoSPARC v4.2.1[74]. Briefly, particles were automatically picked, extracted with a 100 pixel box size, and subjected to 2D classifications into 100 classes. The best class averages were used as templates to re-pick the images, yielding more higher-quality particles. After several rounds of 2D classification, the most informative side view was selected for comparison.

### Proteasome activity assays

Proteasome activity assays used the fluorogenic Suc-LLVY-AMC substrate (MediChem Express) whose cleavage was quantified by the release of free AMC monitored by fluorescence emission at 460 nm following excitation at 365 nm[43]. For each reaction, 100-250 μL of APF

was added to 400 μL of assay buffer containing 100 μM Suc-LLVY-AMC with or without the addition of 0.02% SDS. Reactions were incubated for 20 min at 37 °C and quenched with 500 μL of 80 mM sodium acetate (pH 4.3). The proteasome inhibitors BTZ (1R-3-methyl-1-2S-3-phenyl-2-pyrazin-2-carbonylamino propanoyl amino butyl boronic acid; SelleckChem), MG132 (N-(benzyloxycarbonyl)-leucinyl-leucinyl-leucinal; SelleckChem), and Epo (N-Acetyl-N-methyl-L-isoleucyl-L-isoleucyl-N-(1S)-3-methyl-1-(2 R)-2-methyloxiranyl carbonyl]butyl-L-threoninamide) were obtained Sigma Aldrich. The general protease inhibitors pepstatin-A, bestatin, leupeptin, PMSF, E64 separately, and the plant protease inhibitor cocktail (PIC; containing AEBSF (4-benzenesulfonyl fluoride hydrochloride), aprotinin, bestatin, E64, leupeptin, and pepstatin-A) were purchase from Sigma Aldrich. The CDC48 inhibitor CB-5083 was obtained from MediChem Express.

To quantify the effects of pH on proteasome activity, 100 μL of APFp or CL were mixed with 400 μL of a broad-range pH buffer containing 20 mM Na-citrate, 20 mM $Na_2PO_4$, 20 mM Tris, 2 mM $CaCl_2$, 10 mM NaCl, and 1 mM ATP with its pH adjusted from 4 to 9 with HCl or NaOH. The samples were made 100 μM Suc-LLVY-AMC with or without the addition of 50 μM BTZ, and incubated for 20 min at 37 °C following by quenching with 500 μL of 80 mM sodium acetate (pH 4.3). Levels of released AMC were assayed by fluorescence as above.

## Immunological analyses

For sample preparation prior to immunoblotting, 40 μL of resuspended APFp were heated to 95 °C for 5 min in 10 μL of 5 × SDS sample buffer (250-mM Tris–HCl (pH 6.8), 8% (w/v) sodium dodecyl sulfate, 40% (v/v) glycerol, 20% (v/v) 2-mercaptoethanol, and 0.004% (w/v) bromophenol blue). For the CL, 100 mg of frozen tissue was pulverized and extracted in 300 μL of protein extraction buffer containing 150 mM NaCl, 50 mM Tris-HCl (pH 7.5), 0.1% (v/v) Nonidet P-40, 1% (w/v) 2,2′-dipyridyldisulfide, and 1% plant protease inhibitor cocktail (Sigma Aldrich); the resulting homogenate was clarified at 7000 X*g* for 10 min at 4 °C. A 20-μL aliquot was heated at 95 °C with 5 μL of 5 x SDS sample buffer.

Following SDS-PAGE, the CL and APF proteins were transferred onto Immobilon-P polyvinylidene difluoride (PVDF) membranes (Roche), and the membranes were washed with PBS (137 mM NaCl, 2.7 mM KCl, 10 mM $Na_2HPO_4$, and 1.8 mM $KH_2PO_4$) and blocked overnight with PBS containing 1% (w/v) non-fat dry milk. The membranes were incubated at room temperature with primary antibody solutions (PBS with 1% (w/v) non-fat dry milk) for 60 min, before being washed once with PBS, once with PBS containing 0.1% (v/v) Triton X-100, and once with PBS for 10 min each. The membranes were re-blocked with PBS containing 10% (w/v) non-fat dry milk for 30 min, incubated for 60 min with secondary antibody solution (PBS and 1% (w/v) non-fat dry milk), and then washed again as described above. Primary antisera used at the indicated dilutions were: anti-PAG1 (1:1000), anti-PBA1 (1:1000), anti-RPN1a (1:1000), anti-RPN3 (1:1000), anti-RPN5 (1:1000), anti-PBF1 (1:1000), and anti-rabbit Ub (1:1000) described by ref. [35], anti-Rubisco large subunit (1:3000) (Agrisera-AS03037), anti-cFBP (1:3000) (Agrisera-AS03037), anti-GFP (1:1000) (Abcam ab1218), and anti-actin (1:5000) (Agrisera- AS04043). Secondary antibodies were either the goat anti-mouse HRP conjugate (1:5000–10,000; SeraCare, product number 0741806) or the goat anti-rabbit HRP conjugate (1:5000–10,000; SeraCare, product number 0741506). After a final wash in PBS, proteins were visualized using Super Signal West Pico PLUS chemiluminescent substrate (ThermoFischer Scientific) and imaged with a ChemiDoc Imaging System and/or X-ray film.

For immune-depletion studies related to CP activity, 10 ml of the APFp was resuspended in 500 μL of buffer A (50 mM HEPES (pH 7.5), 50 mM NaCl, 10% (v/v) glycerol, 10 mM $MgCl_2$, 20 mM ATP, 2 mM PMSF, and 0.6% (w/v) $Na_2S_2O_5$). Sigma anti-FLAG M2 resin (200 μL) was washed three times in 500 μL of buffer A, with the beads collected each time by centrifugation 8000 X*g* for 1 min at 4 °C to remove excess

liquid. The washed beads (250 μL) were added at 4 °C to PolyPrep chromatography column, excess liquid was drained, and the APFp was applied three times. The third flow-through was designated as the ex-proteasome-supernatant fraction. The beads were washed three times with 500 μL of Buffer A, collected, and finally assayed as the immunoprecipitated fraction. Equal volumes of the APFp, supernatant, and immunoprecipitated fractions from *PAG1-FLAG pag1-1* leaves were assayed for proteasome activity with the Suc-LLVY-AMC substrate, or subjected to SDS-PAGE and immunoblot analysis as above.

For immune-depletion studies related to ROS bursts, APF was extracted from *PAG1-FLAG pag1-1* leaves with 0.1X TBS (2.5 mM Tris-HCl (pH 7.4) and 14 mM NaCl) plus 50 μM ATP, 0.1X PIC, and 25 μM PMSF. Anti-FLAG M2 resin (150 μL), washed three times in 500 μL 0.1X TBS, was added into 1 mL of APF and agitated at 4 °C for 1 hr, and the beads and supernatant were collected by centrifugation as above. The pelleted beads were washed three times with 500 μL of 0.1X TBS plus ATP, PIC, and PMSF, and the ex-proteasomes finally eluted from the beads with 1 mL 0.1X PBS (0.8 mM $Na_2HPO_4$, 0.2 mM $KH_2PO_4$, 14 mM NaCl and 270 μM KCl (pH 7.4)) amended with 25 μM of a 3X FLAG peptide (GLPBIO), 0.1X PIC, and 25 μM PMSF. Equal volumes of the APF, supernatant, and immunoprecipitated fractions were assays for ROS bursts as described above. For the effects of SylA on proteasome activity, purified proteasomes were affinity purified from *PAG1-FLAG pag1-1* leaves using anti-FLAG beads followed by elution with the FLAG peptide as described[35].

## Mass spectrometric analysis

Proteins from the APF, APFp, and CL were precipitated in 4:1:3 (v/v) methanol/chloroform/ water, collected by centrifugation, washed once more with the same mix, and lyophilized to dryness. The precipitates (-80 ug) were resuspended into 100 μL of 8 M urea, and reduced for 1 hr at room temperature with 10 mM dithiothreitol, followed by alkylation with 20 mM iodoacetamide for 1 hr. The reactions were quenched with 20 mM dithiotreitol, diluted with 900 μL of 25 mM ammonium bicarbonate to reduce the urea concentration below 1 M, and digested overnight at 37 °C with sequencing-grade modified porcine trypsin (Promega) at a trypsin:protein ratio of 1:50. The resulting peptides were lyophilized to a final volume of ~200 μL, acidified with 10% trifluoroacetic acid until the pH was below 3.0, and then desalted and concentrated with Pierce C18 tips (ThermoFischer Scientific) according to the manufacturer's instructions. The peptides were eluted in 50 μL of 75% acetonitrile and 0.1% acetic acid, lyophilized, and resuspended in 15 μL of 5% acetonitrile and 0.1% formic acid for LC-MS/MS analysis.

Nano-scale LC separation of the tryptic peptides was performed using a Dionex Ultimate 3000 Rapid Separation system equipped with a 75 μm x 25 cm Acclaim PepMap RSLC C18 column (ThermoFisher Scientific), in combination with a 2-hr linear 4%-to-36% acetonitrile gradient in 0.1% formic acid and a flow rate of 250 nL/min. Eluted peptides were analyzed online by a Q-Exactive Plus spectrometer (ThermoFisher Scientific) in the positive electrospray ionization mode at a capillary voltage of 2.1 kV. Data-dependent acquisition of full MS scans (mass range of 380–1500 m/z) at a resolution of 70,000 was performed, with the automatic gain control (AGC) target set to $3 \times 10^6$, and the maximum fill time set to 200 msec. High-energy collision-induced dissociation fragmentation of the top eight strongest peaks was performed with a normalized collision energy of 28, an intensity threshold of $4 \times 10^4$ counts, and an isolation window of 3.0 m/z; the process excluded precursors that had unassigned or +1 to +7 charge states. MS/MS scans were conducted at a resolution of 17,500, with an AGC target of $2 \times 10^5$ and a maximum fill time of 300 msec.

The resulting MS/MS spectra were analyzed using Proteome Discoverer (version 2.5, ThermoFisher Scientific), which was programmed to search the *A. thaliana* Col-0 proteome database

(Araport11_pep_20220914) downloaded from TAIR (http://www.tair.com/.). Peptides were assigned using SEQUEST HT[75], with search parameters set to assume trypsin digestion with a maximum of 2 missed cleavages, a minimum peptide length of 6, precursor mass tolerances of 10 ppm, and fragment mass tolerances of 0.02 Da. Carbamidomethylation of cysteines was specified as a static modification, while oxidation of methionines and N-terminal acetylation were specified as dynamic modifications. The target false discovery rate (FDR) of 0.01 (strict) was used as validation for peptide-spectral matches (PSMs) and peptides. Proteins containing equivalent peptides that could not be differentiated based on the MS/MS analysis alone were grouped to satisfy the principles of parsimony. Label-free quantification as previously described[76] was performed in Proteome Discoverer with a minimum Quan value threshold of 0.0001 using unique peptides, and '3 Top N' peptides used for area calculation. All genotypes/treatments were analyzed by four biological replicates, each analyzed by two technical replicates. For inclusion in the datasets, the protein had to be detected in at least one technical replicate for each of the four biological replicates. For proteins detected in two technical replicates, the average values were used, while the actual values were used for those detected by only one technical replicate. Abundance values among the four biological replicates were normalized using the combined MS1 ion intensities (represented as e-values) for the entire sample or its compartments, or for all peptides from individual proteins as a semi-quantitative measure.

Volcano plots were calculated in Persus[77] and generated by the Prism software (version 10; GraphPad). Differences in the APF/APFp versus CL based on four biological replicates were calculated by the students t-test (Log$_2$ FC ≥ 1 or ≤-1, $P$-value ≤ 0.05). To further reduce the number of false positives, proteins with an FDR > 0.05 were also excluded. GO analyses were performed using the *Arabidopsis* profile database in g:Profiler V3.10.1[78] as part of the ELIXIR Infrastructure package (http://biit.cs.ut.ee). GO-annotation categories shown here were selected based on their uniqueness, $P$-values of significance, and degrees of completeness.

### Flagellin isolation and APF digestions

*P syringae* pv. *tomato* DC3000 cells were grown on a solid LB medium containing 100 μg/mL rifampicin, which was used to inoculate 1 L of LB liquid medium containing 10 mM MgCl$_2$ and rifampicin. After 2-d growth at 25 °C with shaking, the cells were collected by centrifugation at 7000 X$g$ for 10 min, and resuspended in 300 mL of minimal medium containing 7.6 mM (NH$_4$)$_2$SO$_4$, 1.7 mM MgCl$_2$, and 1.7 mM NaCl, which was supplemented with 10 mM each of mannitol and fructose. The cells were collected after an overnight incubation at 25 °C and washed three times with 20 mM Tris-HCl (pH 8.0). Flagella were separated from the cells by vortexing vigorously for 1 min twice, removing the intact cells by centrifugation at 7000 X $g$ for 10 min, and filtering the supernatant through a 0.45 μm pore-size Acrodisc syringe filter (Pall Corp.)[52]. The filtrate was centrifuged at 100,000 X $g$ for 60 min to collect released flagella, and resuspended in 0.1 M glycine-HCl (pH 2.0) to dissociate the flagellin subunits. The remaining intact flagella were removed by centrifugation at 100,000 X $g$ for 1 hr, and the supernatant containing flagellin was collected and adjusted to pH 7.0 with NaOH. Prior to incubation with APF, the purified flagellin was incubated at 70 °C for 5 min to reinforce dissociation[52]. For APF-mediated digestion of flagellin, 4 μg of purified flagellin were incubated for 0-4 hr at room temperature with 400 μL of APF isolated from wild-type Col-0 leaves with or without the addition of 1X PIC or 62.5 μM BTZ. The reactions were quenched with SDS-PAGE sample buffer before SDS-PAGE.

### ROS burst assays

For ROS burst assays, APF extraction was as described above, except that the extraction buffer used chilled water containing 50 μM ATP.

*Arabidopsis* leaf discs (5-mm diameter) after dissection were pre-incubated in 100 μL water overnight in a 96-well plate, washed with 150 μL of water, and then incubated with 100 μL water mixed with 100 μL of the tested samples containing 30 μg/mL luminol (dissolved in DMSO) and 30 μg/mL horseradish peroxidase[15]. The flg22 peptide (QRLSTGSRINSAKDDAAGLQIA (GenScript-RP19986) derived from *P. aeruginosa* flagellin) was tested directly at 100 nmoles per well. For *P. syringae* pv. *tomato* DC3000 bacteria and flagellin, 2 μL/well of an OD$_{600}$ 0.5 cultures or 0.3 μg/well (~10 pmol/well), respectively, were first incubated with 200 μL/well of APF for 4 hr at room temperature; the mixtures were frozen and concentrated to a half volume by lyophilization and then added to each well. Chemiluminescence was immediately recorded in Relative Light Units (RLUs) at 2 min intervals using an Infinite M200 plate reader (Tecan Trading AG). To study the effects of protease inhibitors on ROS burst, the APF was pre-incubated with the inhibitors for 20 min prior to bacteria, flg22, or flagellin addition. Each treatment/condition was analyzed by four to five biological replicates. Adjusted fluorescence signals (sample minus Mock) used to generate the TLI values were aggregated from timepoints 2 to 40 min.

### Syringolin-A (SylA) synthesis

SylA was synthesized based on routes described by Dai et al.[79]. and Pirrung et al.[80]. starting with Garner aldehyde, and using valinol, (7-azabenzotriazol-1-yloxy)tripyrrolidinophosphonium hexafluorophosphate, "N,N-diisopropylethylamine, Dess Martin periodinane, and tetramethylethylenediamine in the intermediate coupling steps of the macrocycle. The final product at 98% purity was verified by $^1$H-NMR.

### Statistics & reproducibility

All experiments were performed and repeated at least twice independently. LC-MS/MS-based proteomic analysis was performed with four biological replicates using two technical replicates for each biological replicate. Student's t-tests calculated by the Perseus 2.0.11 software were used to analyze the volcano plots. One-way ANOVA with Tukey's tests using Graphpad Prism 10 analyzed the bar chart plots. No statistical method was used to predetermine sample sizes. No data were excluded from the analyses. Plants were grown under the same conditions and located randomly in the growth chambers. Healthy leaves were collected randomly without any bias. Investigators were not blinded to allocation during the experiments and outcome assessments.

### Reporting summary

Further information on research design is available in the Nature Portfolio Reporting Summary linked to this article.

## Data availability

The proteomic datasets generated in this study have been deposited to the ProteomeXchange Consortium via the PRIDE partner repository and are available from the Pride Repository under access code PXD059522. Processed data for Figs. 2 and 4, Table 1, Supplementary Figs. 1, 2 and Supplementary Table 1 are in Supplementary Data. The full description of SylA synthesis can be found online in Supplementary Information. Source data are provided with this paper.

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

## Acknowledgements

We thank Drs. Roger Innes, Brian Rutter, and Benjamin Koch at the Indiana University and Drs. Barbara Kunkel and Cynthia Lee at Washington University for helpful discussions, and Dr. Katherine Basore at the Washington University Center for Cellular Imaging for assistance with the TEM. Thanks also goes to Drs. Julie Bailey-Serres and Adam Book for providing the transgenic *GFP-RLP18B* and *PAG1-FLAG pag1-1* lines, respectively. This work was supported by a grant from General Medical Sciences at the US National Institute of Health (GM-124452) and by funds provided by Washington University in St. Louis to R.D.V.

## Author contributions

H.Z.K., K-E.C., and R.D.V designed the experiments. H.Z.K. developed the APF and APFp purification protocols and conducted their biochemical and TEM characterizations. K-E.C. directed the MS/MS proteomic analyses. M.K. helped with the APF isolations. K-E.C. conducted the ROS burst assays after initial help from H.Z.K. X.G. and J.S provided synthetic SylA. K.E.C. performed the ROS assays with SylA and tested its effects on the APF. R.D.V. wrote the paper with help from the other authors.

## Competing interests

The authors declare no competing interests.
