## [Transparent Peer Review file · Nature Communications]

Proteasomes Accumulate in the Plant Apoplast Where They Participate in Microbe-Associated Molecular Pattern (MAMP)-Triggered Pathogen Defense

Corresponding Author: Professor Richard Vierstra

Version 0:

Reviewer comments:

Reviewer #1

(Remarks to the Author)

The manuscript by Karimi et al describes the identification and validation of the extracellular presence of proteasomes in the plant apoplast under normal growth conditions. The paper is essentially split into two parts, one a proteomic analysis of enriched apoplastic fractions with confirmation of the presence of proteosome in these fractions. The other portion is then the investigation of the potential role of this protein structure during plant defense responses.

The first portion of the paper focuses on the supposed novelty of an “improved method” for isolation of apoplastic fluid from plants. However, there are already numerous published protocols for isolating apoplastic proteins and metabolites, particularly related to plant defense studies, and I can’t discern any substantial differences from these already published methods. Although the authors perform the standard analysis of proteomic samples to argue enrichment/purity, there is still a substantial portion of the sample that represents chloroplast associated proteins. These may well be ‘real’ as there a numerous cases of proteins being in unpredicted or noncanonical locations, but they also may be contaminants from the isolation. Therefore, the first two figures are pretty standard without any arguable novelty; and the section on the proteomic analysis is overly long considering the lack of novelty.

The extension into the confirmation of the extracellular presence of proteasome structures is performed well and is convincing. Previous extracellular proteomic studies in plants have found extracellular ubiquitin and some components of the proteasome, but this is by far the most thorough investigation of the presence of these structures in the plant apoplast. The result is perhaps not surprising considering the existing paradigm of extracellular proteasomes in mammalian studies as well as the previous indications, but it will be a nice addition to the literature.

The biological portion of this work come from the experiments investigating the potential role of the proteosome in liberating MAMPs during a defense response. Although I believe the authors have provided some evidence supporting their hypothesis, there are a number of caveats that interfere with a clean interpretation. First, extracellular ATP is known to serve as a DAMP, triggering many of the same defense responses as MAMPs like flg22. Therefore, I am somewhat suspect that sufficient extracellular ATP naturally exists to allow the proteosome to function under naïve conditions. Second, because the authors add ATP during their APF isolation to maintain the integrity of the proteosome during isolation, they are activating DAMP responses in the background. This preactivation would result in a perturbed system that may or may not accurately reflect the naïve plant conditions under which the plant would encounter invading bacteria. Finally, the use of the proteosome inhibitor using the ROS assay as a readout is problematic. It has been demonstrated that inhibitors such as MG132 interfere with the initial activation of MAMP responses leading to the ROS responses. Therefore, interpretation of the results starts to become muddled because as whether the inhibitors are affecting the production of MAMPs or the response to MAMPs becomes muddled. As with many cases in biological studies, interpretation of inhibitor studies can always be complicated by extraneous effects, which is why genetics are necessary to demonstrate that an activity “is” involved instead of “could be” involved. As such, I believe the current work is possibly correct, but it falls short of clearly demonstrating a role for the extracellular proteosome during normal plant defense responses.

Reviewer #2

(Remarks to the Author)

Zand Karimi et al. report on the protein composition of the Arabidopsis leaf apoplast. They clearly demonstrate that the apoplast contains active proteasomes (ex-proteasomes) and that ex-proteasomes participate in pattern-triggered immunity (PTI) by converting bacterial proteins into microbe-associated molecular patterns (MAMPs). They also provide data supporting the assertion that syringolin A (SylA) inhibits PTI by inhibiting ex-proteasomes. The catalogue of proteins in the apoplast, though somewhat biased (see below), will be of wide interest. Also below are suggestions to strengthen conclusions about the role of ex-proteasomes and SylA in PTI, which will be of high interest to a more focused audience.

Main comments:

1. The authors report on "Proteomic Analysis of the APF". However, following isolation of the apoplast fluid, the material is subjected to a 100K x g spin to enrich for large particles, yielding a fraction referred to as APFp. This procedure was developed for prior studies of extracellular vesicles. Why was this step included if the goal was to catalog all apoplast proteins? The rationale for this step should be provided and the caveat about how it may skew the apoplast proteome should be explicitly discussed. Furthermore, the authors should determine the percentage of total apoplast proteins recovered by this enrichment step to provide a clear view of how much protein is "left behind" in the supernatant.
2. The role of the ex-proteasomes in pattern triggered immunity is nicely shown. However, ex-proteasome-dependent production of Pst-derived MAMPs other than that from processing of flagellin, e.g. of EF-Tu to elf18 (or active elf18-containing fragments), should be further examined. Much of the PTI in response to Pst is mediated by three RLKs; FLS2, EFR, and CERK1. The authors should use available single and polymutants of the genes encoding these receptors to more clearly define the PTI pathway(s) in which ex-proteasomes participate. Notably, since the elicitor of CERK1 may not require proteolytic processing, the *cerk1* mutant could account for background activation by APF + Pst in the presence of PIC and BTZ (e.g., in figure 6d).
3. The data in figure 8c do not adequately demonstrate that SylA inhibits the ROS burst by preventing processing of flagellin in the APF. An alternative possibility is that SylA inhibits the ROS burst by targeting intracellular proteasomes (or by some other unknown activity). To determine if SylA has such an alternate/additional activity, the authors should test the effect of SylA addition to APF with *flg22* (instead of flagellin) in this assay.

Minor comments:

4. Figure 7 does not add much and could be moved to supplement. If it is not, figure 7 is called out ahead of 6h and 6i.
5. Panel 8a and 8B should be switched to match order of call out in text.
6. Line 67 – Rather than "nutrient acquisition", perhaps "nutrient transport".
7. Line 72 – Rather than "secretion", perhaps "transport". Things are moving both directions across PMs.
8. Line 85 – "Catabolic" implies a role in metabolism. Perhaps instead use "proteolytic".
9. Line 500 "breakdown" should be as "break down".

The headers for the supplementary data need fixed.

Current: Supplementary Data 1. Total list of 2,259 Rubisco proteins detected here by MS in the CL.

Proposed: Supplementary Data 1. Total list of 2,259 proteins detected by MS in the CL.

Current: Supplementary Data 2. Total list of 987 proteins detected here by MS in the APFp

Proposed: Supplementary Data 2. Total list of 987 proteins detected by MS in the APFp.

Current: Supplementary Data 3. Total list of 236 proteins detected specifically by MS in the APFp versus CL.

Proposed: Supplementary Data 3. Total list of 236 proteins detected by MS in the APFp but not in the CL.

Reviewer #3

(Remarks to the Author)

This is an interesting manuscript on the presence of proteasomes in the plant apoplast, their role in facilitating the immune response to pathogens, and the counteraction of a pathogen in suppressing this apoplastic proteasome activity. In general, the results convincingly support the main conclusions by the authors. There are however some issues with how some of the data have been interpreted and there are also some contradictory results, possibly because of naming errors.

Main comments:

1) My main concern is with the analyses of the apoplast proteomics data and the comparisons with the intracellular proteome. The authors have struggled with the presence of chloroplastic proteins (especially rubisco) and some others, in the apoplastic fluid (APF) which prevents or complicates subsequent GO analyses to determine the success rate of their extraction. To solve this problem, the authors have removed proteins that "do not belong in the APF" according to previous results from the scientific community. This is a form of hypothesis driven data exclusion, which is unscientific. This removal of proteins that were extracted as part of the APF may also be misleading as they indeed could be bona fide APF components. For example, the chlorophyll result suggests strongly that there was no leakage of chloroplast constituents into the APF. It is logical to assume that chlorophyll as a much smaller molecule than rubisco, would be more readily leaking into the APF when released from chloroplasts due to the extraction procedure. Therefore, the near absence of chlorophyll in the APF, suggests that the more abundant presence of rubisco is due to other factors than chloroplast leakage.

To be sure, the authors have convinced me that they have isolated the APF and that proteasome particles are enriched in

the APFp, the main conclusion that supports the remainder of the work. Perhaps, this beginning part of the results section could be rephrased with some statements suggesting that the removed proteins could indeed be part of the APF, but that their inclusion in further proteomic analyses complicates comparisons with prior work?

2) There appears to be a contradiction in Figure 4 related to the relative abundance of proteasome subunits in the APFp versus the CL. In the main text, it is stated that:

P15: "Preference for the CP versus RP in the APFp was also seen when comparing relative protein abundances as calculated by combined MS1 ion intensities. Whereas the levels of the CP subunits were collectively higher in the APFp versus CL, those for RP subunits were collectively lower (Fig. 4c)."

This is contradicted by Fig 4b. Figure 4b also contradicts Figures 4d and e?

According to Fig 4b, there is no enrichment of the CP in the APFp compared to the CL. Also, see Figure 5d and the PBA1 subunit.

Minor comments:

1) P13: "By comparison, another large particle possibly in the APFp might have been Rubisco, but its 16-subunit, more ellipsoid barrel of 8 large and 8 small subunits are intimately entwined without obvious tiers."

So, this rubisco particle was not observed?

2) P13: "Less frequently, we also detected structures resembling the CP singly capped by the asymmetric RP, and in rare situations, we detected CPs doubly capped with two RPs_{29,40} (Figs. 3b and 3c), whose assemblies were presumably enforced by ATP added to the APF extraction buffer."

What are the frequencies, and do they correspond with the results shown in Figure 4? This is not essential information, but if the results are available, it would be good to include them.

3) The authors found that autophagy is likely not involved in the trafficking of proteasomes towards the apoplast. This should not be surprising as these ex-proteasomes do not contain the RPN10 subunit that serves as the autophagic receptor for proteasomes. Perhaps this should be mentioned in the manuscript?

4) The ex-proteasomes differ from CL proteasomes in that they contain different isomers for the CP gate subunits PAA and PAE. As the CP is the main proteasome version in the apoplast, could it be that these different alpha subunits provide a different type of gate functioning?

5) P17: "As shown in Figs. 5c and 5d, anti-FLAG beads effectively removed both the CP peptidase activity and the CP subunits PAG1(7) and PBA1(1) and the RP subunit RPN1 from the APFp, with the ex-proteasome activity, CP subunits, and FLAG now enriched in the bound fraction."

This is a bit of an overstatement, there is still significant activity in the supernatant (Fig 5c).

6) P18: "A burst comparable to flg22 was seen for the *P. syringae* + APF samples, while no burst was seen with the APF alone and only a modest burst was seen with cells alone (Fig. 6d)."

Why would the cells alone display a burst while no burst was observed with the APF-alone treatment?

Version 1:

Reviewer comments:

Reviewer #1

(Remarks to the Author)

The authors have been very professional in both addressing a few items of experimentation as well as editorial changes. Together, I believe that they have done a nice job of addressing my previous concerns; and these additions provide greater clarity regarding the interpretation of the results. While some caveats likely still remain about the absolute purity of the apoplastic fractions (always an issue) and some of the biology of the PAMP responses, I feel that they have largely addressed most issues within the limitations of the experimentation (i.e. I cannot think of a way to more clearly demonstrate their main conclusions other than what they have performed).

I feel that - even with a few remaining caveats - this work deserves publication as it is the most complete work addressing an additional paradigm involved in plant-microbe interactions. There may well be some refinement of the models in time, but these results will move the field forward (and by extension will likely stimulate future investigations into other possible roles for extracellular proteasomal activity).

Reviewer #2

(Remarks to the Author)

The authors have satisfactorily addressed my concerns and, in my opinion, those raised by the other reviewers.

Reviewer #3

(Remarks to the Author)

The authors have addressed all my concerns.

Attached please find our revised manuscript entitled "Proteasomes Accumulate in the Plant Apoplast where They Participate in Microbe-Associated Molecular Pattern (MAMP)-Triggered Pathogen Defense" by Hana Zand-Karimi, Kuo-En Chen, Marilee Karinshak, Xilin Gu, Jason Sello and me for possible publication in *Nature Communications* (NCOMMS-24-48926). We were pleased to read such favorable reviews of the original manuscript. In the revised draft we hopefully covered most, if not all, of the recommendations/concerns made by Reviewers.

In the narrative below, the Reviewers' comments are in italic, our responses are in normal type, and additions to the text are in bold type. Highlighted in yellow are Lines in the text where substantial changes were made to the revised version to address the Reviews' concerns. Most of these corrections can be found by yellow highlights in a marked-up version of the manuscript.

REVIEWER COMMENTS

Reviewer #1

(Remarks to the Authors):

The manuscript by Karimi et al describes the identification and validation of the extracellular presence of proteasomes in the plant apoplast under normal growth conditions. The paper is essentially split into two parts, one a proteomic analysis of enriched apoplastic fractions with confirmation of the presence of proteosome in these fractions. The other portion is then the investigation of the potential role of this protein structure during plant defense responses.

The first portion of the paper focuses on the supposed novelty of an "improved method" for isolation of apoplastic fluid from plants. However, there are already numerous published protocols for isolating apoplastic proteins and metabolites, particularly related to plant defense studies, and I can't discern any substantial differences from these already published methods. Although the authors perform the standard analysis of proteomic samples to argue enrichment/purity, there is still a substantial portion of the sample that represents chloroplast associated proteins. These may well be 'real' as there are numerous cases of proteins being in unpredicted or noncanonical locations, but they also may be contaminants from the isolation. Therefore, the first two figures are pretty standard without any arguable novelty; and the section on the proteomic analysis is overly long considering the lack of novelty.

The extension into the confirmation of the extracellular presence of proteasome structures is performed well and is convincing. Previous extracellular proteomic studies in plants have found extracellular ubiquitin and some components of the proteasome, but this is by far the most thorough investigation of the presence of these structures in the plant apoplast. The result is perhaps not surprising considering the existing paradigm of extracellular proteasomes in mammalian studies as well as the previous indications, but it will be a nice addition to the literature.

Authors Response

We agree that the apoplastic proteomics will be a great addition to the literature. In writing the paper, we felt that we needed to describe our isolation method and its results in detail, since its purity underpins our conclusions that proteasomes can be found in the apoplast and not just from cytosolic contamination. While there were several prior attempts at describing the proteomics of the apoplast in the literature, none really went to great lengths to describe how they considered their preparations relatively pure. We think that our simple strategy of holding the leaves up-side-down with the petioles up was likely a key feature in preventing cytoplasm and phloem from oozing out of the cut ends and contaminating the preparations.

We also agree with Reviewer #1 that we went to great lengths to support the notion that proteasome can be found outside of plant cells, using all our tools at hand.

Reviewer #1 (Remarks to the Authors):

The biological portion of this work come from the experiments investigating the potential role of the proteosome in liberating MAMPs during a defense response. Although I believe the authors have provided some evidence supporting their hypothesis, there are a number of caveats that interfere with a clean interpretation. First, extracellular ATP is known to serve as a DAMP, triggering many of the same defense responses as MAMPs like flg22. Therefore, I am somewhat suspect that sufficient extracellular ATP naturally exists to allow the proteosome to function under naïve conditions. Second, because the authors add ATP during their APF isolation to maintain the integrity of the proteosome during isolation, they are activating DAMP responses in the background. This preactivation would result in a perturbed system that may or may not accurately reflect the naïve plant conditions under which the plant would encounter invading bacteria. Finally, the use of the proteosome inhibitor using the ROS assay as a readout is problematic. It has been demonstrated that inhibitors such as MG132 interfere with the initial activation of MAMP responses leading to the ROS responses. Therefore, interpretation of the results starts to become muddled because as whether the inhibitors are affecting the production of MAMPs or the response to MAMPs becomes muddled. As with many cases in biological studies, interpretation of inhibitor studies can always be complicated by extraneous effects, which is why genetics are necessary to demonstrate that an activity “is” involved instead of “could be” involved. As such, I believe the current work is possibly correct, but it falls short of clearly demonstrating a role for the extracellular proteosome during normal plant defense responses.

Authors Remarks:

Reviewer #1 is right that extracellular ATP can serve as a signal for cellular damage via Damage-Associated Molecular Patterns (DAMP) such as ATP working through purinoceptors on the plasma membrane, and could trigger the ROS burst responses seen here with *P. syringae* cells, flagellin, or the 22-amino-acid long peptide flg22. We initially did not consider that this effect was possible with our protocols as the APF prepared with ATP (0.05 or 1 mM) in this study did not elicit a ROS burst when added to leaf disc assay alone. To make this conclusion more clear, we tested the effects of various concentrations of ATP mixed with the APF in the ROS burst assays triggered by flg22. As can now be seen in new Supplementary Figure 5, ATP alone did not trigger a ROS burst when concentrations from 5 to 500 μ M of ATP was added to the APF, and concentrations of ATP from 5 to 50 μ M did not suppress the ROS burst triggered by flg22 when added to the APF. Only 500 μ M of added ATP was problematic, so we kept the ATP concentrations at 50 μ M or lower for the assays.

The notion proposed by Reviewer #1 that the proteasome inhibitors used here could be suppressing MAMP triggered ROS burst does not appear to fit our data as well. As shown in revised Figure 6b,c and revised Figure 8b,c (formally found in Supplementary Data), none of the proteasome inhibitors suppressed the ROS response triggered by flg22 alone but did so for the ROS response triggered by flagellin. This differential effect thus places the effects of the inhibitors as being before the downstream response to flg22 and after the upstream treatment of the discs with flagellin. Moreover, as flg22 perception takes place at the plasma membrane via the FLS2 flg22 pathogen recognition receptor, it would also be reasonable to conclude that the effects of the inhibitors would be outside of the cell.

We agree with Reviewer #1 that basing our conclusions solely on the efficacy of proteasome inhibitors could be potentially problematic. Consequently our alternative approach to implicate proteasomes in the flagellin-mediated MAMP response in the original draft of the

paper was via immune-depletion assays. As previously shown in Figures 5c,d and Figures 6h,i, we used a transgenic line expressing a complementing PAG1-FLAG construction that tagged the CP complex of the 26S proteasome by modifying the $\alpha 6$ subunit. Thus, by simple anti-FLAG bead pull downs, we showed that that we could both remove proteasome activity specifically from the APF and dampen the ability of the APF to trigger a flagellin-triggered ROS burst. See Lines 442-449 and Lines 585-597 in the Results for the description of these immune-depletion studies. We hope that Reviewer #1 agrees with us that this alternative approach does help support our conclusions first based on inhibitor studies.

Reviewer #2
(Remarks to the Authors):

Zand Karimi et al. report on the protein composition of the Arabidopsis leaf apoplast. They clearly demonstrate that the apoplast contains active proteasomes (ex-proteasomes) and that ex-proteasomes participate in pattern-triggered immunity (PTI) by converting bacterial proteins into microbe-associated molecular patterns (MAMPs). They also provide data supporting the assertion that syringolin A (SylA) inhibits PTI by inhibiting ex-proteasomes. The catalogue of proteins in the apoplast, though somewhat biased (see below), will be of wide interest. Also below are suggestions to strengthen conclusions about the role of ex-proteasomes and SylA in PTI, which will be of high interest to a more focused audience.

Main comments:

1. The authors report on “Proteomic Analysis of the APF”. However, following isolation of the apoplast fluid, the material is subjected to a 100K x g spin to enrich for large particles, yielding a fraction referred to as APFp. This procedure was developed for prior studies of extracellular vesicles. Why was this step included if the goal was to catalog all apoplast proteins? The rationale for this step should be provided and the caveat about how it may skew the apoplast proteome should be explicitly discussed. Furthermore, the authors should determine the percentage of total apoplast proteins recovered by this enrichment step to provide a clear view of how much protein is “left behind” in the supernatant.

Authors Remarks:

We agree that it would have been helpful to include a proteomic analysis of the APF prior to centrifugal enrichment. Consequently, we have now added such data in the revised manuscript, which can be found in Figure 2, Table 1, and the Supplementary Data 1-3 files. We thank Reviewer #2 for convincing us to do this experiment. We moved the proteomic analysis of the pelleted APF fraction (APFp) to the Supplementary Data to save space (Suppl. Fig 2 for example). The proteomic analysis of the APF and APFp were quite similar and generated the same conclusions that the APF is enriched in apoplastic proteins, especially those connected to plant defense (e.g., BGAL1, subtilase SBT1.7 and SBT5.2) along with some chloroplast contamination such as Rubisco. We did search for proteins preferentially found in the APF versus APFp, but found that most of the differences were not likely meaningful due to the stochastic nature of identifications by LS-MSMS.

Reviewer #2 (Remarks to the Authors):

2. The role of the ex-proteasomes in pattern triggered immunity is nicely shown. However, ex-proteasome-dependent production of Pst-derived MAMPs other than that from processing of flagellin, e.g. of EF-Tu to elf18 (or active elf18-containing fragments), should be further examined. Much of the PTI in response to Pst is mediated by three RLKs; FLS2, EFR, and CERK1. The authors should use available single and polymutants of the genes encoding these

receptors to more clearly define the PTI pathway(s) in which ex-proteosomes participate. Notably, since the elicitor of CERK1 may not require proteolytic processing, the *cerk1* mutant could account for background activation by APF + Pst in the presence of PIC and BTZ (e.g., in figure 6d).

Authors Response:

We agree that it would be nice to try other protein-based MAMPs. Unfortunately, these experiments will take considerable time as we don't have the MAMPs available even to start the assays. Consequently, as the flagellin/flg22 system is one of the best understood MAMP responses and used as a paradigm in a number of ground-breaking studies, we hope that Reviewer #2 will agree that our data with flagellin/flg22 will at least provide sufficient evidence that more work will be helpful in connecting ex-proteosomes to MAMP signaling writ large. In addition, adding more studies on other MAMPs will probably exceed the space and figure restrictions of *Nature Communications*.

Reviewer #2 (Remarks to the Authors):

3. The data in figure 8c do not adequately demonstrate that *SylA* inhibits the ROS burst by preventing processing of flagellin in the APF. An alternative possibility is that *SylA* inhibits the ROS burst by targeting intracellular proteosomes (or by some other unknown activity). To determine if *SylA* has such an alternate/additional activity, the authors should test the effect of *SylA* addition to APF with flg22 (instead of flagellin) in this assay.

Authors Response:

Reviewer #2 might have missed this point, but the original draft of the manuscript already had these experiments showing that *SylA* does not dampen the ROS burst response to flg22 but does so for flagellin. To make these data more visible, we moved them from the Supplementary Data to panels 8c and 8d in a reorganized Figure 8.

Reviewer #2 (Remarks to the Authors):

Minor comments:

4. Figure 7 does not add much and could be moved to supplement. If it is not, figure 7 is called out ahead of 6h and 6i.

We personally like Figure 7 in the main body of the paper because it shows the effectiveness of each inhibitor by IC50 values, which are consistent with the reported KI values for the drugs.

5. Panel 8a and 8B should be switched to match order of call out in text.

Done as recommended.

6. Line 67 – Rather than “nutrient acquisition”, perhaps “nutrient transport”.

Done as recommended.

7. Line 72 – Rather than “secretion”, perhaps “transport”. Things are moving both directions across PMs.

Done as recommended.

8. Line 85 – “Catabolic” implies a role in metabolism. Perhaps instead use “proteolytic”.

We prefer to keep “catabolic” as both proteases and other hydrolases are included in the sentences

9. Line 500 “breakdown” should be as “break down”.
Done as recommended.

The headers for the supplementary data need fixed.

Current: Supplementary Data 1. Total list of 2,259 Rubisco proteins detected here by MS in the CL.

Proposed: Supplementary Data 1. Total list of 2,259 proteins detected by MS in the CL.

Current: Supplementary Data 2. Total list of 987 proteins detected here by MS in the APFp

Proposed: Supplementary Data 2. Total list of 987 proteins detected by MS in the APFp.

Current: Supplementary Data 3. Total list of 236 proteins detected specifically by MS in the APFp versus CL.

Proposed: Supplementary Data 3. Total list of 236 proteins detected by MS in the APFp but not in the CL.

The entire collection of spread sheets in the Supplementary Data has been reorganized to accommodate the new proteomics data on the APF isolated without centrifugal concentration.

Reviewer #3 (Remarks to the Author):

This is an interesting manuscript on the presence of proteasomes in the plant apoplast, their role in facilitating the immune response to pathogens, and the counteraction of a pathogen in suppressing this apoplastic proteasome activity. In general, the results convincingly support the main conclusions by the authors. There are however some issues with how some of the data have been interpreted and there are also some contradictory results, possibly because of naming errors.

Main comments:

1) My main concern is with the analyses of the apoplast proteomics data and the comparisons with the intracellular proteome. The authors have struggled with the presence of chloroplastic proteins (especially rubisco) and some others, in the apoplastic fluid (APF) which prevents or complicates subsequent GO analyses to determine the success rate of their extraction. To solve this problem, the authors have removed proteins that “do not belong in the APF” according to previous results from the scientific community. This is a form of hypothesis driven data exclusion, which is unscientific. This removal of proteins that were extracted as part of the APF may also be misleading as they indeed could be bona fide APF components. For example, the chlorophyll result suggests strongly that there was no leakage of chloroplast constituents into the APF. It is logical to assume that chlorophyll as a much smaller molecule than rubisco, would be more readily leaking into the APF when released from chloroplasts due to the extraction procedure. Therefore, the near absence of chlorophyll in the APF, suggests that the more abundant presence of rubisco is due to other factors than chloroplast leakage. To be sure, the authors have convinced me that they have isolated the APF and that proteasome particles are enriched in the APFp, the main conclusion that supports the remainder of the work. Perhaps, this beginning part of the results section could be rephrased with some statements suggesting that the removed proteins could indeed be part of the APF, but that their inclusion in further proteomic analyses complicates comparisons with prior work?

Authors Response:

We agree that we have not been perfectly clear as to which proteins were assumed to be cytosol contaminants and thus removed from the 3,127 proteins listed as likely residing in the apoplast based on a compilation data from prior MS analyses and GO designations. This predicted apoplast list was then used to help identify which proteins could be assigned to the apoplast in our proteomic datasets. The excluded list was quite short and only included 29 proteins/isoforms that include Rubisco large and small subunits, ribosomal proteins, proteasome subunits, histones, and photosynthetic proteins. They are now listed in Supplementary Data 4 and the text on **Lines 188-191** was amended as shown below. We chose to remove proteasomes as to not bias our enrichment comparisons.

“The composite list was then culled for proteins known or previously predicted to reside in other compartments (e.g., Rubisco and its activase, ribosomal proteins, proteasome subunits, histones, PEP carboxylase, and proteins integral to photosynthetic light capture; 29 in total (see Supplementary Data 4))....”

For additional statistical support in removing Rubisco from the APF and APFp samples as likely contaminants, we compared the variation of the MS1 ion count data measuring Rubisco levels to those of the top 20 proteins in the CL and APF by Pearson’s Correlation Analysis and R² values. For each analysis, the variations in Rubisco levels better correlated with the variations in the CL protein levels than with the APF/APFp protein levels. We added this statement to **Lines 241-244** in the Results to read:

“Further analyses of the MS1 ion-intensity data by Pearson’s correlation and R² values comparing Rubisco levels to those of the top 20 CL and APF proteins were also consistent with Rubisco behaving as a variable contaminant in the APF preparations.”

Reviewer #3 (Remarks to the Author):

2) There appears to be a contradiction in Figure 4 related to the relative abundance of proteasome subunits in the APFp versus the CL. In the main text, it is stated that:

P15: “Preference for the CP versus RP in the APFp was also seen when comparing relative protein abundances as calculated by combined MS1 ion intensities. Whereas the levels of the CP subunits were collectively higher in the APFp versus CL, those for RP subunits were collectively lower (Fig. 4c).”

*This is contradicted by Fig 4b. Figure 4b also contradicts Figures 4d and e?
According to Fig 4b, there is no enrichment of the CP in the APFp compared to the CL. Also, see Figure 5d and the PBA1 subunit.*

Authors Response:

We are unsure as the nature of the statements above. Moreover, the data have also been modified in Figure 4c to include the proteomic analysis of the APF in addition to the APFp in panel. What we generally see is that the abundance of the CP versus RP based on the cumulative MS1 ion counts for all subunits in each complex is generally higher in the APF than the CL. We also see this by immunoblotting CP and RP subunits in the CL versus APFp based in proteasome activity as shown in Figure 4e.

The immunoblot data in Figure 4b is not comparable as the antibodies for the CP and RP subunits tested have strongly different detection strengths.

Minor comments:

1) P13: *“By comparison, another large particle possibly in the APFp might have been Rubisco, but its 16-subunit, more ellipsoid barrel of 8 large and 8 small subunits are intimately entwined without obvious tiers.”*

So, this rubisco particle was not observed?

We did not see obvious particles resembling Rubisco. **Lines 315-318** was amended to read:

“By comparison, another large particle possibly in the APFp might have been Rubisco; however its 16-subunit, more ellipsoid barrel of 8 large and 8 small subunits intimately entwined without obvious tiers⁴² was not obvious in the preparations.”

2) P13: *“Less frequently, we also detected structures resembling the CP singly capped by the asymmetric RP, and in rare situations, we detected CPs doubly capped with two RPs^{29,40} (Figs. 3b and 3c), whose assemblies were presumably enforced by ATP added to the APF extraction buffer.”*

What are the frequencies, and do they correspond with the results shown in Figure 4? This is not essential information, but if the results are available, it would be good to include them.

While these frequency data might have been useful, we did not see enough singly and doubly capped 26S complexes containing both the RP and CP to make the results statistically meaningful. Seeing the RP alone was additionally challenged due to its more diffuse and asymmetric shape. By contrast, the CP was easy to see by its rigid 4-stacked ring barrel shape.

3) *The authors found that autophagy is likely not involved in the trafficking of proteasomes towards the apoplast. This should not be surprising as these ex-proteasomes do not contain the RPN10 subunit that serves as the autophagic receptor for proteasomes. Perhaps this should be mentioned in the manuscript?*

We agree that the absence of RPN10 in the ex-proteasome complexes could be very interesting. We became more confident of this conclusion upon MS/MS of the APF, where we also could not detect RPN10. We added this observation **to Lines 353-357** in the Results and **Lines 812-817** in the Discussion.

“Besides lacking RP13 and RPN14/SEM1, the most conspicuous absence was RPN10 that functions as both a Ub-receptor and an adaptor for the autophagic clearance of proteasomes³⁰. Its known ability to partition between particle-bound and free forms^{46,47} implies that RPN10 could dissociate from the RP within the apoplast or during secretion.”

“The lack of detectable RPN10 in the Arabidopsis leaf APF/APFp, which is needed to stabilize interactions between the Lid and Base subcomplexes of the RP⁶⁹, and the preferential accumulation of the PAA1(α 1) and PAD1(α 4) subunit isoforms in the CP raise the intriguing notion that plant ex-proteasomes are a unique proteolytic complex, possibly analogous to mammalian immunoproteasomes and thymoproteasomes that selectively incorporate unique CP β subunits for basal defense and antigen presentation⁷⁰.”

4) *The ex-proteasomes differ from CL proteasomes in that they contain different isomers for the CP gate subunits PAA and PAE. As the CP is the main proteasome version in the apoplast, could it be that these different alpha subunits provide a different type of gate functioning?*

We agree that this difference seen for PAA and PAE in the APFp could have been interesting. Unfortunately, this isoform specificity was not seen for the APF. Instead, the isoform differences that now stood out were for the PAA1 versus PAA2 and PAD1 and PAD2 in the CL versus APF as shown in Supplementary Figure 1b. This preference is now mentioned in the text on Lines 349-351 in the Results and Lines 812-817 of the Discussion (see above).

“A notable difference was the strong preference for the PAA2 and PAD1 isoforms in the APF and APFp samples based on presence/absence and volcano plots (Fig. 4a; Supplementary Fig. 1b), indicating that the apoplast might harbor a unique CP subtype.”

5) P17: *“As shown in Figs. 5c and 5d, anti-FLAG beads effectively removed both the CP peptidase activity and the CP subunits PAG1(α 7) and PBA1(β 1) and the RP subunit RPN1 from the APFp, with the ex-proteasome activity, CP subunits, and FLAG now enriched in the bound fraction.”*

This is a bit of an overstatement, there is still significant activity in the supernatant (Fig 5c).

We amended this preference on Lines 446-448 to say:

“As shown in Figs. 5c and 5d, anti-FLAG beads significantly removed both the CP peptidase activity and the CP subunits PAG1(α 7) and PBA1(β 1) and the RP subunit RPN1 from the APFp, with the ex-proteasome activity, CP subunits, and FLAG now enriched in the bound fraction.”

6) P18: *“A burst comparable to flg22 was seen for the P. syringae + APF samples, while no burst was seen with the APF alone and only a modest burst was seen with cells alone (Fig. 6d).” Why would the cells alone display a burst while no burst was observed with the APF-alone treatment?*

As we see it, the cells and flagellin alone (without APF pretreatment) do elicit a weak but detectable “ROS signal” as seen in Figure 6d and 6e. It is delayed as compared to APF digested samples with the delay presumably reflecting the time needed for penetration of the inducers into the leaf discs and generation of the MAMP signal (*i.e.*, flg22). We better explained our observations on Lines 529-561 to say:

“Interestingly, we noticed both *P. syringae* cells and purified flagellin alone invariably stimulated modest but protracted ROS responses and TLI values (~44% and 23% relative to with APF, respectively) when added to the leaf discs with fluorescence still evident after 1 hr (Figs. 6d and 6g). We presume that these delayed kinetics reflect the time needed for *P. syringae* cells and flagellin penetration into the leaf apoplast and subsequent cleavage into MAMPs such as flg22 by endogenous ex-proteasomes and/or other peptidases/proteases”.